# No disconnection syndrome after near-complete callosotomy
Selin Bekir [1] ✉, Johanna L. Hopf[2], Theresa Paul[3], Valerie M. Wiemer[3], Tyler Santander[1,4], Henri E. Skinner[1], Anna Rada[2], Friedrich G. Woermann[2,5], Thilo Kalbhenn [2,6], Barry Giesbrecht [1,4], Christian G. Bien[2], Olaf Sporns [7], Michael S. Gazzaniga[1], Lukas J. Volz[3,8] & Michael B. Miller [1,4,8] ✉

Sensorimotor processing in the human brain is largely lateralized, with the corpus callosum integrating these processes into a unified experience. Following complete callosotomy, this integration breaks down, resulting in disconnection syndromes. We asked how much of the corpus callosum is sufficient to support functional unity—the absence of disconnection syndrome—by comparing three complete callosotomy patients with one retaining only the splenium. Using lateralized tasks across visual, tactile, visuospatial, and language domains, we predicted domain-specific deficits in the splenium-only patient based on established anatomical models of callosal topography. Strikingly, while complete callosotomy patients exhibited disconnection syndromes, the splenium patient demonstrated functional unity across all domains—as if his entire corpus callosum were intact. Our findings highlight the brain's remarkable capacity to maintain behavioral integration through minimal preserved pathways, highlighting how the structure-dependent reorganizational capacity of the human brain may allow to preserve functional unity.

When we began testing a new cohort of callosotomy patients, one patient, BT, stood out with unexpected behavioral outcomes during our initial bedside tasks. On tasks that typically reveal inter-hemispheric disconnections, he showed no signs of impairment, performing comparably to neurotypical individuals with an intact corpus callosum. A closer look at his structural MRI scan and the neurosurgeon's report revealed the source of this surprising observation: a small portion of BT's posterior corpus callosum (part of the splenium) was left intact to prevent surgical complications. This seemingly minor detail would go on to challenge our assumptions about the corpus callosum's functional organization and its mechanistic role in integrating information across the human cerebral hemispheres.

The corpus callosum (CC) is the largest white matter tract connecting the left and right cerebral hemispheres[1]. Evidence from callosotomy patients—in whom the CC is surgically severed—shows that the CC plays a crucial role in integrating sensorimotor events across the hemispheres. In lateralized tasks that isolate sensory input and motor output to a single hemisphere, these patients show a profound disruption in inter-hemispheric information flow, known as the disconnection syndrome. Although other commissural pathways —such as the anterior, posterior, and hippocampal commissures—typically remain intact and may provide limited residual inter-hemispheric integration[2], they cannot fully compensate for the loss of the corpus callosum, as patients still typically exhibit disconnection syndromes[3]. For example, consider a callosotomy patient whose speech areas are located in the left hemisphere. When the word "hot" is presented in their left visual field (processed by the right hemisphere) and the word "dog" in their right visual field (processed by the left hemisphere), they verbally report seeing only the word "dog". However, when asked to draw what they saw using their left hand (controlled by the right hemisphere), they drew a fire. Crucially, the emergent concept of a hotdog is never reported, even though each hemisphere accurately processes its respective input[4,5]. This demonstrates the lack of integration and awareness of sensorimotor events in the opposite hemisphere when the CC is fully severed. By contrast, neurotypical individuals with an intact CC do not show such disconnection effects, exhibiting what we refer to as 'functional unity'. We define functional unity as the absence of a disconnection syndrome—where lateralized inputs and outputs are readily shared and integrated across hemispheres, enabling unified perception and behavior.

The CC is thought to be topographically organized, where anterior callosal fibers project to anterior cortical regions like the prefrontal cortex,

[1]Department of Psychological & Brain Sciences, University of California, Santa Barbara, Santa Barbara, CA, USA. [2]Department of Epileptology, Krankenhaus Mara, Bethel Epilepsy Center, Medical School OWL, Bielefeld University, Bielefeld, Germany. [3]Medical Faculty, University of Cologne, and Department of Neurology, University Hospital Cologne, Cologne, Germany. [4]Institute for Collaborative Biotechnologies, University of California, Santa Barbara, Santa Barbara, CA, USA. [5]MRI Department, Society for Epilepsy Research, Krankenhaus Mara, Bethel Epilepsy Center, Bielefeld, Germany. [6]Department of Neurosurgery, Evangelisches Klinikum Bethel, Medical School OWL, Bielefeld University, Bielefeld, Germany. [7]Department of Psychological & Brain Sciences, Indiana University, Bloomington, IN, USA. [8]These authors contributed equally: Lukas J. Volz, Michael B. Miller. ✉e-mail: sbekir@ucsb.edu; mbmiller@ucsb.edu

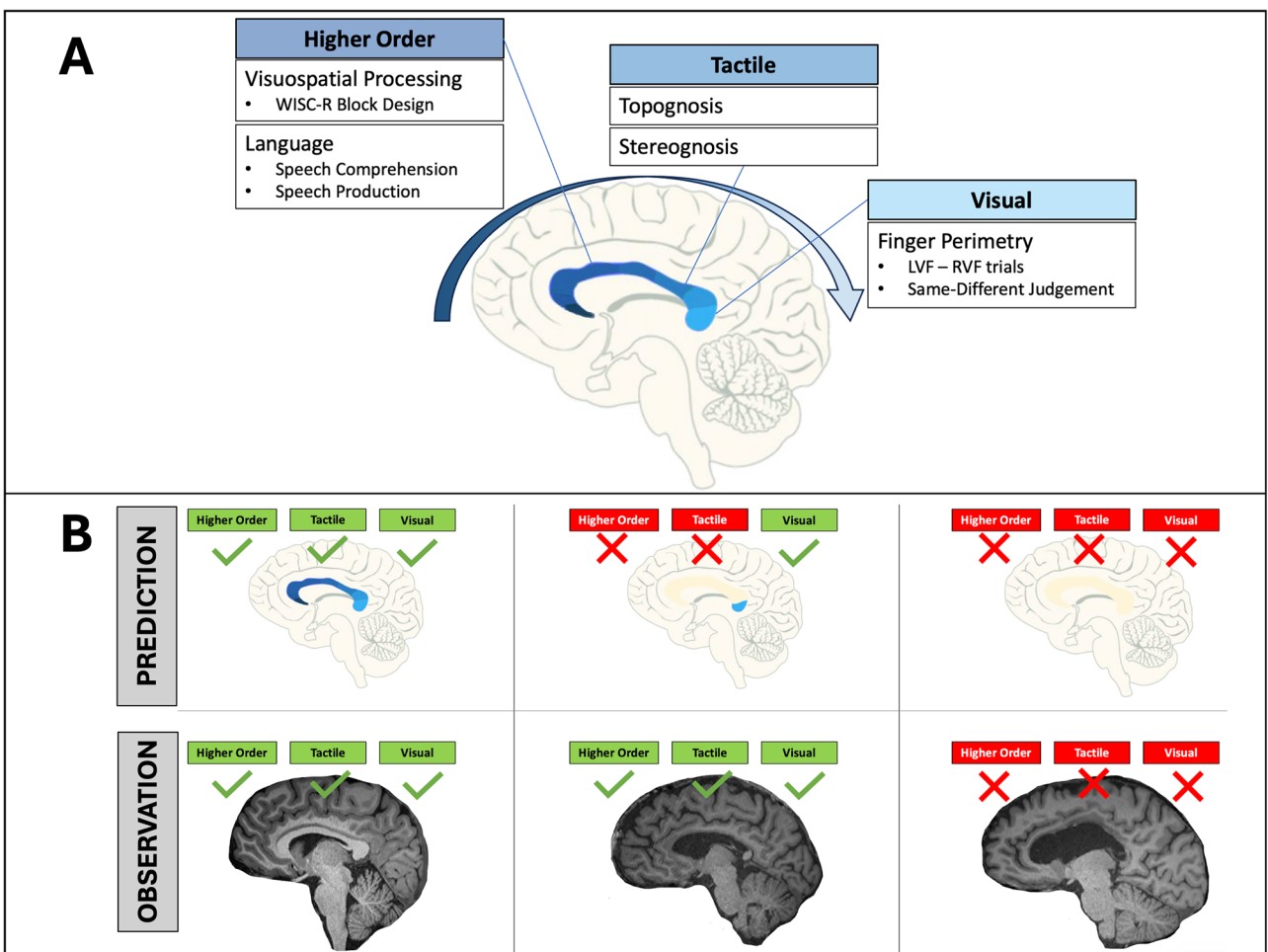

**Fig. 1 | Overview of bedside tests, hypothesized corpus callosum (CC) topography, predicted outcomes, and observed results. A** Assumed topographic and modality-specific organization of the CC and our bedside tests. The splenium (the most posterior part of the CC) is typically associated with visual integration, while the posterior body is associated with tactile transfer. We used simple bedside tests to assess each domain: finger perimetry for visual integration; topognosis and stereognosis for tactile; and additional tasks probing higher-order functions (visuospatial and language) to test the impact of hemispheric disconnection. LVF left visual field, RVF right visual field. **B** Predicted (top) vs. observed (bottom) disconnections across three schematic cases: a neurotypical brain (intact CC, shown for reference only), BT* (with only the splenium intact), and a complete callosotomy (patient TJ). Green checks indicate preserved integration; red crosses indicate disconnection. Based on CC topography, we predicted that BT* would show visual integration only. However, BT* exhibited intact integration across all domains, suggesting preserved inter-hemispheric integration despite very limited callosal connectivity.

and posterior fibers project to posterior regions like the occipital lobe[6]. As a result, the CC is argued to have a modality-specific organization with different subsections—from anterior to posterior: rostrum, genu, body, isthmus, and splenium—facilitating information integration in distinct cognitive and sensorimotor domains[7,8]. The splenium, the most posterior section of the CC, is typically implicated in visual integration, while the posterior midbody is thought to be involved in tactile integration[9–11]. Evidence from partial callosotomy cases sparing these posterior subregions supports this notion[7,12–15]. However, in most previous partial callosotomy patients, both the splenium and posterior midbody remain intact, rendering a precise functional distinction impossible. The question thus arises: which kinds of information can the splenium alone integrate across hemispheres?

Cases in which only the splenium is left intact are incredibly scarce. Most callosotomy procedures are either completed in a single stage (severing the entire CC) or performed in two stages, with the first stage preserving more than just the splenium[16]. Only a few splenium-only cases exist in the literature, and their findings are mixed: some suggest the splenium supports broad inter-hemispheric integration[17], while others indicate it is insufficient beyond visual information[18]. However, these studies have limitations—some lack MRI confirmation of exactly preserved structures[17], while others conducted testing very soon after surgery (within 6 months), making it

unclear whether any reorganization had occurred over time[18]. As a result, the specific contribution of the splenium to inter-hemispheric integration remains unclear.

The current study addresses this gap by reporting on a rare, splenium-only callosotomy case, compared against three complete callosotomy patients. We used an array of lateralized bedside tasks to sample performance across the anteroposterior axis of the CC to identify disconnection syndromes in visual, tactile, and higher-order cognitive functions to determine where performance breakdowns occur after complete versus partial callosotomy, (see Fig. 1A). Based on the framework of the CC's topographic and modality-specific organization, we predicted that complete callosotomy would result in a full disconnection syndrome. Conversely, the partial callosotomy patient with only ~1 cm of splenium intact was expected to show disconnection effects in all tasks except the visual modality, reflecting restricted inter-hemispheric integration along splenial fibers connecting the bilateral visual cortex (see Fig. 1B).

## Methods
### Participants
We tested four patients (BT, NR, TJ, LJ) who underwent callosotomy to alleviate epileptic seizures. Three patients (NR, TJ, LJ) had complete

**Table 1 | Clinical profiles of callosotomy patients**

| Patient | Age at surgery | Age at testing | Time from surgery in months | Sex | Handedness | Intelligence | Extent of callosotomy | Education (years) |
|---|---|---|---|---|---|---|---|---|
| BT* | 22 | 28 | 70+ | M | Right | Low average | Partial | 10 years |
| TJ | 49 | 52 | 34 | F | Right | Average | Complete | 13 years |
| LJ | 57 | 60 | 26 | M | Right | Low average | Complete | 10 years |
| NR | 18 | 18 | 6 | M | Right | Moderately impaired | Complete | 12 years |

Patients Overview. +Patient BT was first assessed 70 months after surgery, showing no disconnections; a full battery was conducted later (~87 months post-surgery), yielding the same results. Patient Intelligence classifications are based on the Stanford–Binet Fifth Edition (SB5) classification[48,49]. Sex refers to biological sex and was determined from participants' medical records.

**Fig. 2 | Anatomical scans of the patients.** Midsagittal T1-weighted MRI images with overlaid DTI-based fiber tractography are shown for each patient. Only select commissural fibers—the anterior commissure and corpus callosum –as well as some cerebellar and brainstem fibers, are displayed. Tractography parameters were adjusted for each patient to provide the clearest midsagittal depiction. The fiber orientations are color-coded: red for left-right, green for anterior-posterior, and blue for superior-inferior directions. All patients exhibit an intact anterior commissure (white arrow), but only patient BT* retains a preserved splenium (yellow arrow), in contrast to the complete sectioning of the corpus callosum in the remaining three patients.

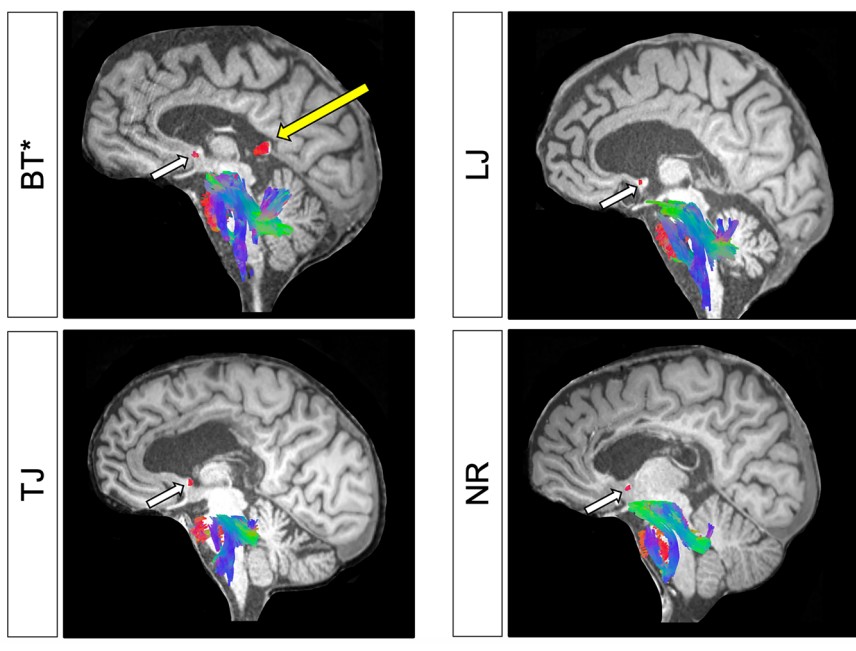

callosotomy, which fully sectioned the corpus callosum. In BT, only posterior fibers were left intact due to technical surgical reasons. All patients were tested post-operatively (see Table 1 and Fig. 2). The surgeries effectively reduced drop seizures in most patients. TJ and NR were entirely seizure-free post-surgery (100% reduction), BT experienced a near-complete reduction (98%), and LJ showed moderate improvement with a 20% decrease in drop seizures. No sedation was administered during behavioral testing or MRI scanning procedures. All reported patients here are right-handed[19], and have language areas in the left hemisphere, confirmed by pre-operative bilateral Wada tests[20]. All patients completed secondary school degrees (10–13 years of education). Post-operative general neuropsychological testing indicated performance ranging from below average to average on working memory and executive function tasks (WAIS-IV Digit Span, forward and backward)[21], with BT and NR performing in the below-average range and TJ and LJ falling in the low-average to average range. Dominant-hand praxis was intact in all patients, as assessed by the Apraxia Screen of Tulia (AST)[22]. For ease of comprehension, BT is referred to as BT* to denote his status as a partial callosotomy patient. Finally, the anterior, posterior, and hippocampal commissures remained intact in all patients. The study was approved by the Ethics Committee Westphalia-Lippe (2021-523-f-S). Each patient or the legal representative consented to participate in the study as approved by the Ethics committee. Patients received €100 for attending the examination program at the Bethel Epilepsy Center. Travel, accommodation, and meal expenses were reimbursed with an average of €250 per person. Partial compensation was provided to those who missed some examinations.

### Bedside testing of the disconnection syndrome

We conducted a total of five bedside tests to sample performance along the anteroposterior axis of the corpus callosum (see Fig. 1). The tasks included visual, tactile, language, and visuospatial tasks, described in detail below. Critically, the tasks comprised an 'intra-hemispheric' (uncrossed) condition, where the input and output hemispheres were the same, and an 'inter-hemispheric' (crossed) condition, where the input and output hemispheres differed. We operationalized the presence of a disconnection syndrome as (i) above-chance performance in *intra-hemispheric* conditions with concurrent (ii) chance-level performance in *inter-hemispheric* conditions for a given task. This contrast establishes that the patient can understand and perform a given task when it is confined to a single hemisphere (*intra-hemispheric* uncrossed trials) and *only* fails to do so if it requires interhemispheric integration (*inter-hemispheric* crossed trials). Please see Fig. 3 for the details of the tasks and conditions. There was no preregistration for this study.

### Visual tasks

**Finger perimetry.** The experimenter showed varying numbers of fingers in the patient's peripheral vision (in the left or right visual field; LVF or RVF) while the patient fixated on the experimenter's nose. Then the patient verbally reported the number of fingers while keeping their gaze fixated at the experimenter (i.e., center of the visual field). A same/different version of the task was also conducted: The experimenter showed fingers on both sides simultaneously while the patient maintained central fixation and reported whether the numbers were the same or different.

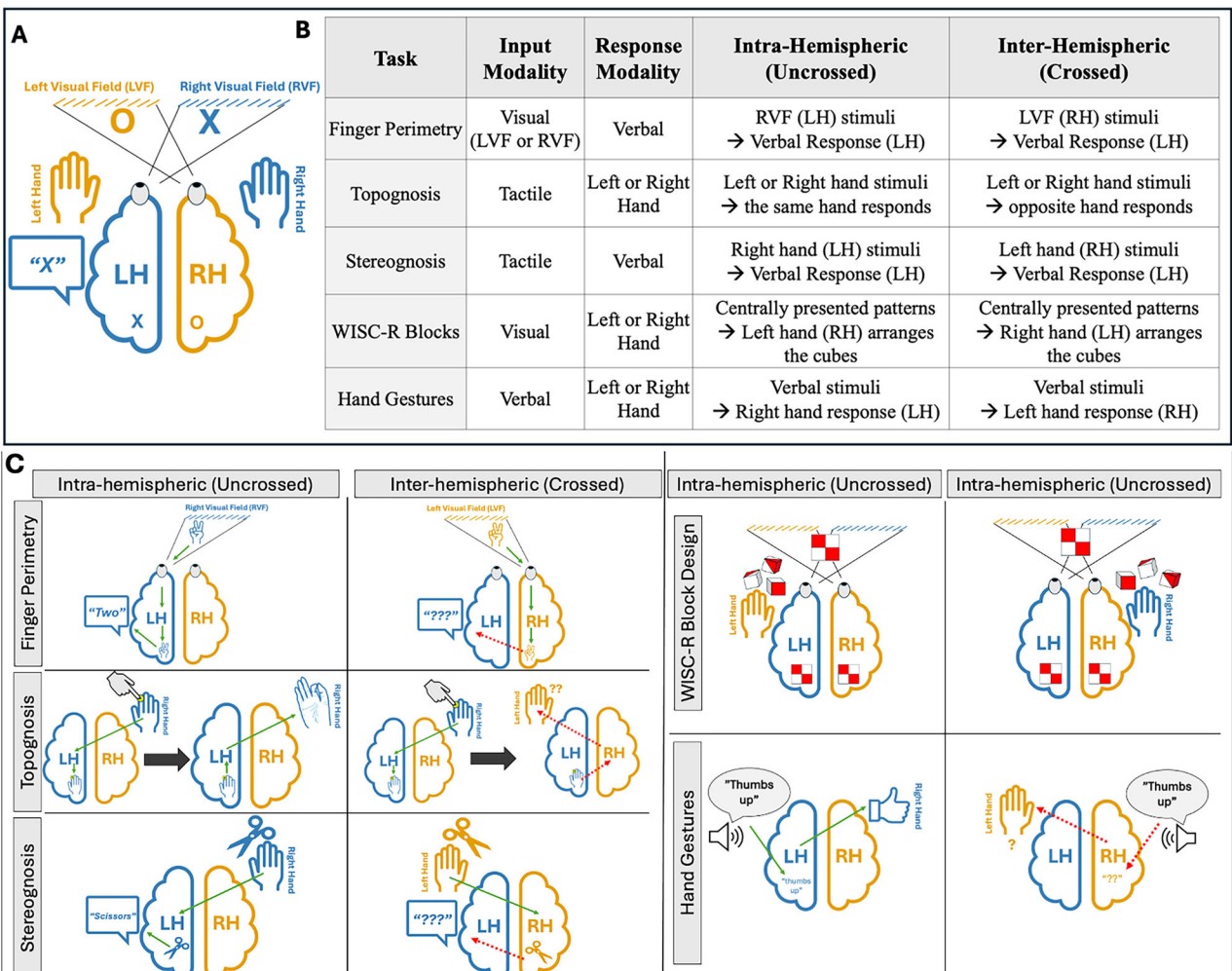

**Fig. 3 | Classic split-brain framework and task descriptions. A** Hemispheric organization of inputs and outputs. Stimuli in the left visual field (LVF) projects to the right hemisphere (RH), while stimuli in the right visual field (RVF) projects to the left hemisphere (LH). Motor outputs are also lateralized: the LH controls speech and the right hand, while the RH controls the left hand. **B** Task summary table showing, for each task, the input and the response modality, and examples of uncrossed and crossed trial types. **C** Examples illustrating uncrossed and crossed trials for each task.

**Inter- and intra-hemispheric trials.** Since all patients have speech areas in the left hemisphere and this task required verbal responses, RVF trials (processed by the left hemisphere) were intra-hemispheric trials, and LVF trials were inter-hemispheric trials. In the same/different task, all trials were inter-hemispheric in nature.

**Chance-level performance.** It was 20%, as patients could guess any number from one to five in each trial. In the same/different version of the task, the chance level was 50% when guessing "same" or "different".Due to LJ's difficulties with maintaining fixation and/or motivation to perform the task, finger perimetry was successfully conducted for BT*, TJ, and NR only.

**Tactile tasks**
**Topognosis—intra- and inter-manual touch localization.** The experimenter lightly touched specific points on the fingers of the blindfolded patient's right or left hand. In the intra-manual (i.e., *intra-hemispheric*) condition, the patient used their thumb of the same hand to indicate the stimulated spot; in the inter-manual (i.e., *inter-hemispheric*) condition, the patient used their thumb of the opposite hand to show the corresponding spot on the opposite hand. For example, if the tip of the right index finger was touched, the patient pointed to the tip of the left

index finger with their left thumb. Alternatively, in some cases, the patient rested their hands on the table. After the experimenter lightly touched the distal segments of the fingers, they indicated their location by raising the corresponding finger. Both versions measure the same underlying ability—tactile localization in intra- and inter-manual conditions. Moreover, the tasks were always compared within subjects—contrasting intra-manual and inter-manual conditions for the same patient—and the task procedure remained consistent across both intra- and inter-manual conditions within each subject. Thus, the findings remained comparable across conditions, and the chance levels were adjusted according to the specific procedure used for each patient.

**Inter- and intra-hemispheric trials.** *Intra-hemispheric* trials were the summed intra-manual trials from the left and right hands separately; *inter-hemispheric* trials were inter-manual trials summed across both directions (left-hand stimulated, right hand responds, and vice versa).

**Chance-level performance.** For patients NR and TJ, who completed the task using their thumb (indicating tactile stimulation across four fingers), the chance level was 25%. For patients BT and LJ, who completed the task by raising one of their five fingers to indicate the location, the chance level was 20%.

**Stereognosis—naming objects held in hand**. Blindfolded patients were given common objects (e.g., scissors, spoon, fork, ball) to identify using tactile exploration with one hand at a time, alternating between the left and right hand on separate trials. Then, they were asked to name the presented objects. A towel was used to prevent any auditory cues.

**Inter- and intra-hemispheric trials**. Since all patients have speech areas in the left hemisphere and this task required verbal responses, right-hand trials (processed by the left hemisphere) were *intra-hemispheric*, and left-hand trials were *inter-hemispheric*.

**Chance-level performance**. It was 20% as there were five trials (five objects) per hand.

### Hemispheric lateralization of visuospatial processing

**WISC-R block design**. The patient was given colored cubes with surfaces that were solid red, solid white, or half red and half white bisected across the diagonal, and was shown pictures of patterns. Patients were then asked to replicate these patterns using the cubes[23]. A total of 16 trials were conducted, with eight trials using the right hand and eight using the left hand. The first two trials involved creating simpler patterns using two cubes, while the remaining six trials involved arranging four cubes into 2 × 2 square patterns.

**Inter- and intra-hemispheric trials**. Previous studies suggest a left-hand (right-hemisphere) dominance over the right-hand trials[24]. Thus, left-hand (right-hemisphere) trials were considered *intra-* and the right-hand (left-hemisphere) trials were considered *inter-hemispheric* trials.

**Chance-level performance**. It was not clearly defined for this task. Thus, the left-hand (intra-hemispheric) and the right-hand (inter-hemispheric) trials were contrasted to assess the disconnection syndrome.

### Hemispheric lateralization of language

Language is a classic example of a lateralized cognitive function[25,26]. In most people, the language network is mainly located in the left hemisphere[27,28]. Thus, for all language tasks, *intra-hemispheric* corresponded to right-hand response trials (controlled by the left hemisphere) and *inter-hemispheric* corresponded to left-hand response trials (controlled by the right hemisphere).

**Language comprehension in the form of speech: hand gestures task**. The patient performed hand gestures with either the left or right hand, following verbal instructions, with their eyes closed. The gestures included thumbs up, thumbs down, an okay sign, a fist, and two or four fingers up.

**Chance-level performance**. It was not clearly defined for this task. Right-hand trials (*intra-hemispheric*) and left-hand trials (*inter-hemispheric*) were contrasted to assess the disconnection syndrome.

**Control—mimicking task**. We included a control task to rule out the possibility of general motor impairments, motivational issues, or a broader inability of the right hemisphere to perform a given task using the left hand—independent of its language processing capabilities. In this task, patients, with their eyes open, were asked to mimic hand gestures demonstrated by the experimenter using their left hand. The ability to successfully mimic the gestures in this control condition but not when following verbal instructions suggested that the deficit was not due to motor impairments, lack of motivation, or general performance difficulties, but rather a difficulty in understanding the verbal command. Therefore, performance on this control task was more relevant for the cases where one or both hands failed to perform the main task (i.e., executing hand gestures following verbal commands). In such cases, good

performance in this control task could serve to rule out general motor deficits or motivational issues as potential explanations for poor performance in the main task.

**Language production in the form of speech**. Tasks that require a verbal response to lateralized stimuli (e.g., finger perimetry and stereognosis) were used to assess speech production from the left hemisphere and the right hemisphere. While we previously noted that all patients were left hemisphere dominant for speech production, as determined by the Wada protocol prior to surgery, there are reports in the literature suggesting that sometimes the non-dominant right hemisphere can develop the capacity to generate speech years after surgery[29]. Thus, assessing and reporting speech production from both hemispheres is important during periodic post-operative assessments.

### Statistical analyses

We conducted binomial tests (two-tailed) to determine whether the patient's performance differed from chance-level performance for *inter-* and *intra-hemispheric* trials. The disconnection syndrome was operationalized as (i) above-chance performance on *intra-hemispheric* trials and (ii) chance-level performance on *inter-hemispheric* trials for the same task. Significance ($p < 0.05$) indicated that the observed performance was statistically different from chance-level; non-significance ($p \geq 0.05$) indicated that the observed performance did not differ from chance. To account for the fact that all tests were conducted in a clinical setting with a limited number of trials, we also performed Monte Carlo simulations with 10,000 iterations to ensure the robustness of our findings. We generated new data under the null hypothesis (chance-level performance) via simulations to estimate the distribution of outcomes and determine the significance of the observed empirical results. For all tasks, the Monte Carlo simulations produced results consistent with the binomial tests.

For tasks where chance performance was not well-defined (WISC-R Block Design and hand gestures), we used permutation tests to compare performance between the right and left hands. To assess this, we used the absolute difference between the hands' proportion-correct scores as our test statistic. The permutation test was implemented with 10,000 iterations. For each iteration, we first created binary arrays representing correct (1) and incorrect (0) trials for each hand. We then pooled all these trials from both hands and randomly reassigned them to either hand while maintaining the original number of trials per hand. For each such permutation, we calculated the absolute difference in proportion correct between hands. The *p*-value was computed as the proportion of permuted differences that were equal to or greater than the observed difference. This approach allowed us to estimate how likely it would be to observe a difference as large as the one we found if there were no differences between the hands (statistical significance was set at $p < 0.05$).

Lastly, despite the limited trial numbers and small sample size, our statistical inferences were based on within-subject comparisons of *intra-* versus *inter-hemispheric* trials within each task. This approach allowed each patient to serve as their own control. In other words, we used each patient's intra-hemispheric trials as a baseline to compare their inter-hemispheric performance and characterize the disconnection syndrome.

### Disconnection score

This *Disconnection Score* was calculated as the proportion-correct in the *intra-hemispheric* condition (ranging from 0 to 1) minus the proportion-correct in the *inter-hemispheric* condition. We computed the proportion-correct scores from the number of correct trials out of the total trials for each condition (Table 2). For example, if a patient has a proportion correct of 0.9 in the *intra-hemispheric* condition and 0.1 in the *inter-hemispheric* condition, their disconnection score for that task would be 0.8 (calculated as 0.9−0.1). Below is a detailed explanation of the methods used to create the Disconnection Score for each cognitive domain and sensory modality.

**Article**

## Table 2 | Individual performance on disconnection assessments

| Domain | Task | Condition | Patients | | | |
|---|---|---|---|---|---|---|
| | | | BT* | NR | TJ | LJ |
| *Visual* | *Finger perimetry* | *Intra (RVF)* | **4/4** $P = 0.002$ | **4/5** $P = 0.007$ | **3/3** $P = 0.008$ | NA |
| | | *Inter (LVF)* | **4/4** $P = 0.002$ | 2/5 $P = 0.263$ | 0/3 $P = 1$ | NA |
| | *Same-Diff.* | *Inter-* | **9/10** $P = 0.021$ | 4/10 $P = 0.754$ | NA | 2/4 $P = 1$ |
| *Tactile* | *Topognosis* | *Intra-manual* | **10/10** $P < 0.001$ | **7/8** $P < 0.001$ | **26/26** $P < 0.001$ | **3/4** $P = 0.027$ |
| | | *Inter-manual* | **10/10** $P < 0.001$ | 3/8 $P = 0.422$ | 4/30 $P = 0.204$ | 0/3 $P = 1$ |
| | *Stereognosis* | *Intra (right hand)* | **5/5** $P < 0.001$ | **5/5** $P < 0.001$ | **6/6** $P < 0.001$ | **4/5** $P = 0.007$ |
| | | *Inter (left hand)* | **4/5** $P = 0.007$ | 3/5 $P = 0.058$ | 0/1 | 1/5 $P = 1$ |
| Visuospatial | Block design | Intra (left hand) | 8/8 | 8/8 | 5/6 | NA |
| | | Inter (right hand) | 8/8 $P = 1$ | 8/8 $P = 1$ | 0/2 $P = 0.105$ | NA |
| Language | Hand gestures | Intra (right hand) | 6/6 | 6/6 | 3/3 | 3/4 |
| | | Inter (left hand) | 6/6 $P = 1$ | 5/6 $P = 1$ | 4/12 $P = 0.076$ | 0/3 $P = 0.14$ |

Performance of each patient across tasks and conditions. The table presents the number of correct trials for each patient. Tasks in *italic* (e.g., Visual and Tactile tasks) were analyzed using binomial tests to assess whether performance exceeded chance-level. Bold scores indicate above-chance performance ($p < 0.05$), while non-bold scores reflect chance-level performance. Notably, BT* consistently performed at ceiling levels in both *inter-* and *intra-hemispheric* conditions. In contrast, complete callosotomy patients generally struggled with *inter-hemispheric* tasks but performed well on *intra-hemispheric* trials. Block Design and Hand Gesture tasks were analyzed using permutation tests to compare the left- and right-hand performances, with a single *p*-value reported for each comparison. While BT* and NR showed no differences between hands, TJ and LJ exhibited trends suggestive of differences. However, due to limited and unmatched trial numbers (e.g., 5/6 vs. 0/2 for TJ), none of the comparisons were statistically significant (all $p > 0.05$). Some tasks were aborted early (e.g., 0/2) when patients showed visible struggles, to prevent further frustration. These methodological limitations are discussed in the "Results" and "Discussion" sections.

**Visual.** The proportion of correct trials in intra-hemispheric (right visual field) trials minus inter-hemispheric (left visual field) trials from Finger Perimetry was used for the visual disconnection score.

**Tactile.** Topognosis (intra-manual minus inter-manual trials) was used to calculate the tactile disconnection score, which is considered one of the most sensitive tests for assessing tactile disconnections[30].

**Speech production.** For BT*, NR, and TJ, we used Finger Perimetry (RVF minus LVF trials) and Stereognosis (right-hand minus left-hand trials). Difference scores were calculated and averaged for each task. For LJ, only Stereognosis data was available, so the difference score was based solely on this task.

We only included disconnection scores for tasks with clearly defined chance levels and binomial test outcomes. WISC-R spatial block design and hand gestures tasks were not included here because the combination of limited trial numbers and undefined chance levels significantly reduced our statistical power, making the statistical detection of disconnection syndrome challenging in these tasks, even though qualitative evidence supports their presence (see "Results" section).

### Diffusion MRI acquisition and fiber tractography
To generate tractography images for illustration purposes (Fig. 2), diffusion-weighted images were acquired on a 3T Siemens Vida at the Bethel Epilepsy Center using a 32-channel head coil. A single-shot echo-planar imaging sequence was used with one $b = 0$ volume, and diffusion-encoding gradients applied along 30 directions at $b = 1000$ s/mm² (TR = 13.2 s, TE = 83 ms, flip angle = 90°, slice thickness = 2 mm, voxel size = $2 \times 2 \times 2$ mm³, FOV = 256 mm). Parallel imaging was applied using GRAPPA (acceleration factor = 2).

Tractography was performed in DSI Studio[31]. Diffusion data were reconstructed using generalized q-sampling imaging (GQI)[32], and deterministic streamline tracking was applied[33]. The anisotropy threshold was randomly selected. The step size was set to voxel spacing. Parameters such as the angular threshold (35–90° range), and the total number of seeds were placed (around 50,000 seeds) were individually adjusted across patients to achieve optimal anatomical visualization. Tracks shorter than 40 mm or longer than 200 mm were excluded. Tractography was used solely for illustrative purposes in Fig. 2 and was not analyzed quantitatively. Only select commissural, cerebellar, and brainstem fibers were retained for display after manual pruning to enhance clarity. Each tractography rendering was overlaid on the patient's own T1-weighted structural image to highlight the anatomical landmarks relevant to surgical outcomes.

### Reporting summary
Further information on research design is available in the Nature Portfolio Reporting Summary linked to this article.

## Results
As predicted, all complete callosotomy patients exhibited signs of a disconnection syndrome. However, contrary to our expectations, the partial callosotomy patient, BT*, who underwent a near-complete resection, only sparing his splenium, showed no behavioral disconnection effects in any of the tested domains (see Table 2).

Across all intra-hemispheric (uncrossed) conditions, patients performed significantly above chance (overall $p < 0.05$; individual $p$-values per task are reported below). These results indicate intact sensory and motor integrity when tasks did not require inter-hemispheric information integration, whereas performance typically broke down once such integration was necessary.

The number of trials varied across patients and tasks because of practical bedside constraints, including fatigue, limited testing time, and stopping some inter-hemispheric trials early to avoid frustration when patients struggled with task performance. For some tasks, when testing time was limited, we prioritized inter-hemispheric trials, once intra-hemispheric ones were performed easily, which led to uneven trial numbers. These case-specific reasons are detailed below; nevertheless, the overall pattern of intact intra-hemispheric and impaired inter-hemispheric performance was generally consistent across patients.

### Visual Integration is preserved with splenial sparing
As predicted, BT* with his intact splenium showed no disconnections in visual tasks. He performed significantly above chance in both RVF and LVF trials (4/4 correct, $p = 0.002$ for both) and accurately identified whether numbers in his periphery were the same or different (9/10 correct, $p = 0.021$). The complete callosotomy cases, NR and TJ, on the other hand, showed expected disconnections. They could correctly report the numbers shown in RVF trials (4/5 and 3/3, respectively, $p < 0.05$ for both), but not in LVF trials (2/5 and 0/3, $p > 0.05$ for both). NR was also unable to report whether the numbers simultaneously presented in the LVF and RFV were the same or different. (4/10, $p = 0.754$).

https://doi.org/10.1038/s44271-025-00377-5 **Article**

### Tactile Integration is preserved with splenial sparing

In the Topognosis (touch localization) task, BT*, despite only having a small portion of the splenium intact, correctly located tactile stimulation in inter- and intra-manual trials (10/10, $p < 0.001$ for both). In contrast to BT*, all complete callosotomy patients (NR, TJ, and LJ) exhibited clear signs of disconnection syndrome. They all failed to cross-locate touch in inter-manual trials (3/8, 4/30, and 0/3, respectively, all $p > 0.05$) but they correctly did so in intra-manual trials (7/8, 26/26, and 3/4, all $p < 0.05$).

In stereognosis task, where we asked patients to name common objects placed in their left or right hand, to test whether tactile information could be transferred to the speaking left hemisphere, contrary to our prediction, patient BT* correctly named the objects handed to his left hand (4/5) and right hand (5/5, $p < 0.05$ for both), showing no signs of disconnection. In contrast to BT*, all complete callosotomy patients (NR, TJ and LJ) failed to name the objects placed in their left hands (3/5, 0/1, and 1/5, all $p > 0.05$), but were able to correctly name them in their right hands (5/5, 6/6, and 4/5, all $p < 0.05$), showing the expected disconnection syndrome.

Note that with TJ, only one left-hand trial could be conducted in this task because she verbally reported feeling nothing on the left hand. Thus, the remaining trials were abandoned to prevent further frustration. While this limited the statistical analysis, it is, in-and-of-itself, an excellent demonstration of the disconnection syndrome: When an object was placed in her left hand—processed by the mute right hemisphere—her disconnected speaking left hemisphere repeatedly insisted she felt nothing. However, note that, during the tactile localization task reported above, she performed perfectly fine with her left hand—accurately locating touch stimuli within her left hand in the intra-manual trials. We administered a higher number of trials in the tactile localization (Topognosis) task for TJ compared to the other patients, specifically to confirm that she was able to perceive and localize the tactile input on the left hand—which she did. This indicates that she did not have any actual sensory loss in her left hand. Instead, it highlights the disconnection between the hemispheres: her right hemisphere could perceive and process tactile input, but her speaking left hemisphere could not access or verbalize that information.

### Right-hemisphere dominance in visuospatial processing after complete callosotomy

To assess the visuospatial disconnection, we used the WISC-R Block design task[23]. BT* showed no disconnection in this task. He correctly completed all patterns with his left and right hands (observed difference in proportion correct = 0).

TJ showed clear disruption when using her dominant right hand but not her left hand, consistent with previous reports[24]. Although only two trials were attempted with her right hand—limiting statistical analysis—anecdotal evidence highlights her disconnection, and potentially the involvement of the left-hemisphere interpreter[34]. TJ visibly struggled with her right hand. She was incapable of arranging the cubes into a simple $2 \times 2$ square, instead repeatedly forming a $3 \times 1$ shape, let alone creating the correct patterns. After some effort, she claimed there was something wrong with her vision. However, when switched to her left hand, TJ quickly and accurately created most of the patterns without difficulty (she completed one as a mirror image, resulting in a score of 5/6). Although this is an anecdotal observation and lacks quantification, it is compelling: her left hemisphere verbally rationalized her right-hand struggles, incorrectly attributing them to vision problems, only for her left hand (mute right hemisphere) to perform the task perfectly. Despite this observation, the permutation test showed no statistically significant difference between the hands (observed difference in proportion correct = 0.83, $p = 0.1$), potentially due to the limited trial numbers with the right hand (0/2).

NR, being a complete callosotomy patient, successfully completed this task with both hands. There were no significant differences between hands (observed difference in proportion correct = 0). His family reported regular rehabilitation exercises involving similar tasks, which may explain his unexpected right-hand performance compared to other complete callosotomy patients in this study and in the literature[24]. Alternatively, it is also

plausible that he possesses bilateral representations for visuospatial processing.

### Speech comprehension and production from the disconnected hemispheres

**Speech comprehension–hand gestures task.** BT* successfully carried out hand gestures following verbal commands with his right hand (6/6) and left hand (6/6), showing no difference between the two hands (observed difference in proportion correct = 0).

In contrast, TJ showed difficulty performing hand gestures with her left hand (4/12) when instructed verbally. Remarkably, she easily and accurately performed all gestures when mimicking the experimenter's hand movements with her left hand (9/9), showing no signs of motor or motivational deficits. The difference between her left-hand performance following verbal commands (4/12) and when simply mimicking the experimenter's hand gestures (9/9) was significant (observed difference in proportion correct = 0.66, $p = 0.004$). This suggests that her right hemisphere may be limited in its ability to comprehend speech. There was, however, no significant difference between her left-hand trials (4/12) and right-hand trials (3/3) following verbal commands (observed difference in proportion correct = 0.66, $p = 0.07$). This is likely due to the small and unmatched number of trials despite the observation that her right hand performed the task without difficulty (3/3), while the left hand struggled (4/12). Similarly, even though LJ performed the task well with his right hand (3/4) and struggled with his left hand (0/3), there was no statistical difference between the hands (observed difference in proportion correct = 0.75, $p = 0.15$). NR, in contrast, had no difficulty performing hand gestures with his left and right hands following verbal instructions (4/5 and 5/5, observed difference in proportion correct = 0.16, $p = 1$), suggesting that he might have bilateral representations for speech comprehension.

**Speech production.** BT* was able to verbally respond to all inputs, whether presented to the left or right hemisphere. In contrast, all complete callosotomy cases demonstrated a clear disconnection syndrome, suggesting that speech production is still exclusively controlled by the left hemisphere. They could verbally respond only to inputs presented to their left hemisphere (e.g., RVF trials in the visual task and right-hand trials in the tactile task that require naming the objects), but not to their right hemisphere (e.g., LVF trials in the visual task and left-hand trials in the tactile naming task).

### A comprehensive disconnection score

To summarize the disconnection syndrome across different domains for each patient, a difference score was calculated between intra- and inter-hemispheric conditions, providing a measure of the degree of disconnection (see Fig. 4). This *Disconnection Score* was calculated as the proportion-correct trials in the *intra-hemispheric* condition (ranging from 0 to 1) minus the proportion-correct trials in the *inter-hemispheric* condition. A Disconnection Score close to 0 indicated no disconnection (i.e., equal performance in intra- and inter-hemispheric conditions); and a Disconnection Score close to 1 indicated full disconnection. Complete callosotomy cases showed varying degrees of severity of the disconnection syndrome across domains, but all exhibited clear signs highlighting a disruption of inter-hemispheric information integration. In contrast, BT* consistently had scores near 0, suggesting no difference between intra- and inter-hemispheric performance—indicating no disconnection syndrome.

## Discussion

This study presents the first behavioral findings from a new cohort of adult callosotomy patients tested between 1 and 6 years post-surgery. We found that the complete callosotomy patients exhibited disconnection syndromes across multiple sensory and cognitive domains. In contrast, the patient with a partial callosotomy, sparing only the splenium, showed no evidence of disconnection syndrome. These results offer two key contributions to our understanding of the cognitive and behavioral effects of hemispheric severance.

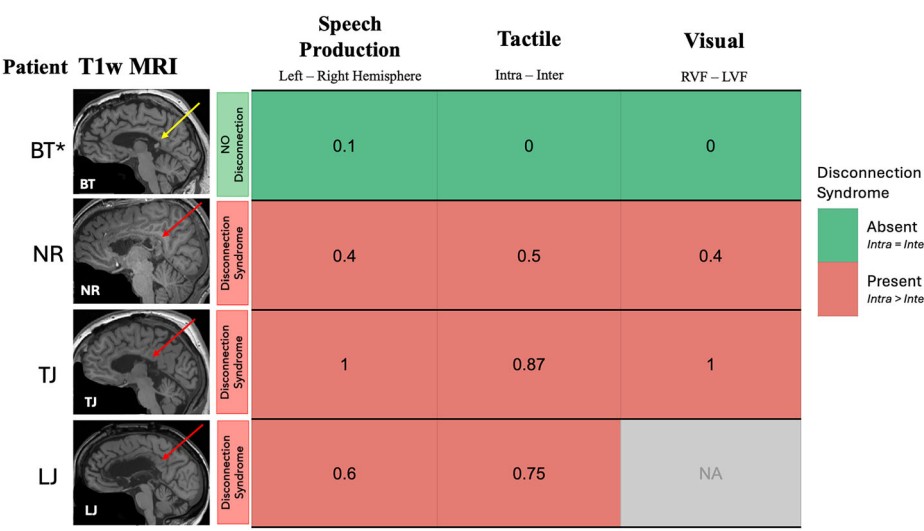

**Fig. 4 | Disconnection scores.** Binary color-coded representation of the disconnection syndrome. *Red* indicates the presence of the disconnection syndrome, defined by above-chance performance in intra-hemispheric trials ($p < 0.05$) and chance-level performance in inter-hemispheric trials. *Green* indicates no disconnection, with above-chance performance in both inter- and intra-hemispheric conditions. *Gray* indicates no available data. Disconnection Scores range from 0 (equal performance in inter- and intra-hemispheric conditions) to 1 (perfect intra-hemispheric performance with chance-level inter-hemispheric accuracy). Structural MRIs with red arrows indicate complete callosotomies, while the yellow arrow indicates a partial callosotomy. Patient BT*, while structurally more similar to complete callosotomy cases with the majority of the CC severed, exhibited no disconnection syndrome, performing equally well in inter- and intra-hemispheric conditions.

**Fig. 5 | Case comparisons: splenium patients.** Overview of previous partial callosotomy cases similar to BT*, with testing times since surgery, observed selective disconnections, and potential factors explaining differences in disconnection profiles. **A** Patient BT* (current study). **B** Case 9[18]. **C** Patient DB[12]. **D** Patient JKN[43]. "Topog." refers to topognosis, "Stereog." to stereognosis. "No" indicates the absence of disconnection syndrome. For case JKN, the testing time is probably within the first year ([†]Prof. Gazzaniga, personal communication). **For case DB, stereognosis performance is unclear. DB was described as showing disconnection syndrome in stereognosis; however, the report states that she successfully named objects with her left hand 75% of the time[12]. This relatively high success rate suggests that her tactile processing may be more similar to BT*'s preserved abilities than to the clear deficits typically seen in complete callosotomy patients.

First, our current results provide new evidence for the ongoing debate regarding the extent of disconnection syndromes in callosotomy patients[35–40]. This recent debate concerns whether the traditional characterization of the 'split-brain phenomenon'—marked by performance disruptions when input and output hemispheres differ during inter-hemispheric trials—is as robust or widespread as traditionally reported. It was sparked by a report of complete callosotomy patients who could respond to lateralized inputs with either hand, leading the authors to argue that "motor unity" persists despite the complete severance of the CC[36]. In contrast, our findings with this new cohort replicate the classical findings: all complete callosotomy patients exhibited clear disconnection syndromes—performing well on intra-hemispheric trials but poorly on inter-hemispheric ones. Hence, there was no behavioral evidence of motor unity. Our results suggest that alternative explanations, such as the cross-cuing (proposed by Volz and Gazzaniga[38]), may account for the apparent unity reported in Pinto and colleagues[39,40].

Second, our study presents a unique and exceptionally rare splenium case whose performance challenges the assumptions of the current anatomical models of callosal topography. Given that BT* retained only the posterior portion of the splenium, we expected to observe a partial disconnection syndrome, such that visual information integration would remain intact, facilitated by the splenium, while more anterior functions such as tactile integration or higher-order functions would be disrupted (see Fig. 1B). However, BT* showed no disconnections at all, leading to the

question how functional unity can be achieved in light of the minimal structural integrity of the corpus callosum.

One critical factor can be the *timing of the testing since surgery*. There are reports of splenium-only patients who demonstrated selective disconnections when tested shortly after surgery[18] (within 2–6 months; see Fig. 5 for case comparisons). In contrast, we tested BT* six years after his surgery. These six years may have allowed enough time for the brain to undergo profound reorganization, with the splenium serving as the critical anatomical substrate enabling the inter-hemispheric transfer of information for various functional networks[41]. While subcortical structures have also been proposed as alternative routes for such reorganization[39,42], all patients in our cohort featured intact subcortical pathways. Yet only BT* showed functional unity across all tested domains. This suggests that it is the presence of the splenium—rather than intact subcortical pathways alone—may enable functional unity across hemispheres.

Further support for this notion stems from resting-state functional connectivity analyses, which revealed that BT*'s inter-hemispheric connectivity was largely sustained[41]. Various measures of functional network integration resembled that of healthy controls with an intact corpus callosum, in stark contrast to complete callosotomy patients (see Santander et al.[41] for further details). These observations show that even small posterior callosal remnants can sustain widespread inter-hemispheric functional coupling, consistent with his intact behavioral performance.

Additionally, BT*'s unexpected behavioral results, compared to the other splenium-only cases, may be explained by *the extent of CC preservation*, in line with a *threshold effect*. For instance, patient JKN[43] (Fig. 5D) may have had a smaller intact portion of the splenium than BT*, which could explain the selective disconnections observed in him but not in BT*. In other words, JKN's smaller intact portion of the CC may have fallen below the critical threshold needed for reorganization, leading to persistent disconnection symptoms. Of note, threshold effects are common in complex systems, where gradual changes often produce no noticeable impact until a critical point is reached, leading to a sudden state shift[44].

*The direction of the callosal resection*—anterior-to-posterior versus posterior-to-anterior—might also be an important factor[45]. One of the proposed evolutionary origins of the CC is argued to be midline fusion for the sensory cortices[1], with humans heavily relying on sensorimotor processes to interact with the world. Thus, posterior inter-hemispheric integration might better preserve the system's integrity and may allow for more effective compensation than vice versa. What, then, is the role of the anterior CC? Our tasks may have been insensitive to effectively probe this question. While we cannot definitively rule out the absence of *any* disconnection syndromes in BT*, he showed no such signs on our battery of bedside assessments. Moving forward, more rigorous and targeted testing with BT* is needed to better probe the anterior CC's role in cognition.

A further relevant comparison can be made with patients who suffer from non-surgical lesions affecting the CC. Such partial lesions to the CC typically result in selective functional impairments that align with the CC's topographic organization[10]—unlike what we observed in BT*. Previous research has highlighted that behavioral outcomes tend to differ between surgical and non-surgical CC disruptions[17], partly because non-surgical lesions may involve more diffuse, extra-callosal damage, whereas callosotomy targets the CC more precisely. Another key difference could be the temporal progression of these lesions: callosotomy involves a sudden severance of the CC, whereas non-surgical lesions may develop more gradually. Initially, sudden disruption may lead to clearer disconnections compared to gradual lesions, where compensatory mechanisms might develop simultaneously with the lesion. However, over time, the non-progressive nature of callosotomy can offer a stable foundation for network reorganization, potentially resulting in the preserved functional unity we observed in BT* six years after his surgery. A similar distinction can be seen in agenesis of the corpus callosum[46], a non-progressive, congenital absence of the CC. Despite the total lack of inter-hemispheric callosal fibers, individuals with callosal agenesis often do not show the disconnection syndromes observed in complete callosotomy patients—likely due to developmental plasticity that reorganizes function from birth.

Methodologically, our study introduces a key contribution: the Disconnection Score. Split-brain research has long lacked standardized tools and metrics, making it difficult to compare cases across studies. While in-depth domain-specific investigations have provided valuable insights into specific aspects of callosal function, we lack a standardized disconnection profile metric that could be reported for each patient regardless of the study's primary focus. Our multi-domain assessment provides a standardized framework for quantifying disconnection severity. By assessing performances across visual, tactile, and higher-order cognitive domains using simple bedside tests, this method aims to provide an initial "disconnection profile" for each patient. Moreover, the tasks are simple enough for patients with intermediate cognitive impairment to perform.

## Limitations

There are four main limitations of our study. First, the *lack of longitudinal* data prevents us from determining whether BT*'s intact performance reflects an absence of disconnection syndrome all along or its resolution through network reorganization over the years post-surgery. While network reorganization remains the most plausible explanation given the existing reports of selective impairments in other splenium cases[18], we cannot definitively establish BT*'s symptom trajectory. Similarly, we cannot fully rule out alternative factors such as rehabilitation-induced plasticity or

premorbid bilateral representations. However, such explanations are highly unlikely to account for all task domains. For example, while premorbid bilateral organization may support functions with hemispheric dominance (e.g., speech comprehension or visuospatial processing), it is insufficient to explain preserved performance on tasks such as topognosis that strongly rely on inter-hemispheric transfer. Second, while BT* showed no disconnection syndrome on our battery of bedside assessments, our current methodology does not allow us to pinpoint the underlying mechanisms for this functional unity. In other words, the precise nature of what is being integrated remains unclear. While the splenium is traditionally associated with visual information integration, recent evidence suggests its fibers also project to temporal cortices[47]. It is possible that what is transferred may not be the sensory input, but more abstract, conceptual representations. More rigorous behavioral testing, alongside event-related functional neuroimaging and high-resolution diffusion MRI—including tractography analyses to identify cortical termination points—will be necessary to further delineate these mechanisms and clarify the functional contributions of preserved splenial pathways. Third, with respect to our language tasks, we make inferences about speech comprehension and production, but more rigorous testing is needed to evaluate these processes in more detail. Our "speech comprehension" task likely engaged a multi-step process: (1) auditory speech perception, (2) semantic comprehension, and (3) motor planning/execution. The control gesture imitation task rules out step 3—praxis or motor deficits—if the patient was able to accurately mimic gestures on sight. Thus, failure to perform gestures on verbal command, despite intact imitation, points to a disruption in the auditory-to-semantic pathway (steps 1 and/or 2), which our current design cannot further dissociate. Future studies using refined controls, such as more rigorous testing of meaningless gesture imitation or lateralized auditory tasks, may help to further delineate why patients were not able to successfully perform the task (e.g., whether it is related to phonological input vs semantics). Similarly, our speech production tasks would benefit from the addition of a systematic non-verbal response condition (e.g., left-hand responses during finger perimetry) to ensure that observed deficits indeed reflected impaired verbalization rather than failures in visual or sensory processing. Finally, our findings are limited in *generalizability* due to the single-subject nature of BT*'s case and the *limited number of trials* due to clinical testing constraints—which significantly reduced statistical power. Then again, several aspects of our study design strengthen the validity of these findings. The within-subject design offers robust control over individual variability, and the selective disruption observed in complete callosotomy cases—only in inter- but not intra-hemispheric conditions—suggests a targeted impairment in tasks requiring inter-hemispheric transfer or integration, rather than a general deficit. This pattern of selective disruption stands in stark contrast to BT*'s intact performance across similar tasks

In conclusion, our study suggests that even a small, intact posterior callosal remnant can sustain remarkable functional capabilities years after surgery. While BT* is a single case, and the absence of longitudinal data prevents us from determining whether this integration was preserved immediately post-surgery or emerged through network reorganization over time, this observation nevertheless highlights the splenium's remarkable role in such integration. Our current findings support the notion that inter-hemispheric structure-function relationships in the human brain do not follow a linear pattern where behavioral disruptions scale with the extent of structural (callosal) resection. Instead, minimal preserved structural connectivity may maintain broad integration beyond what our classical anatomical models predict.

## Data availability

The data supporting the findings of this study are available on GitHub at https://github.com/selinbekir/splitbrain-disconnection-analysis. The structural MRI scans were not quantitatively analyzed and were included only to illustrate the extent of callosal resection (Fig. 2). Due to existing data-sharing agreements with the clinical site, the full MRI datasets cannot be made publicly available at this stage. However, access can be granted upon

**Article**

request following the establishment of an appropriate data-sharing agreement.

## Code availability
Analysis code is openly available on GitHub at https://github.com/selinbekir/splitbrain-disconnection-analysis.

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

## Acknowledgements

S.B., T.S., H.E.S., B.G., and M.B.M. were supported by the Army Research Office under contract W911NF-19-D-0001 for the Institute for Collaborative Biotechnologies. T.P. and L.J.V. were funded by the German Research Foundation (CRC-1451, Project 431549029). The funders had no role in study design, data collection and analysis, decision to publish, or preparation of the manuscript.

## Author contributions

S.B., J.L.H., T.P., A.R., H.E.S., B.G., C.G.B., O.S., M.S.G., L.J.V., and M.B.M. conceived and planned the experiments. T.P., V.M.W., J.L.H., A.R., F.G.W., C.G.B., and L.J.V. assisted with patient recruitment and were responsible for data collection. T.K. performed the callosotomy surgeries. S.B., T.S., L.J.V., and M.B.M. conceived the data analysis strategy; S.B. contributed to the analysis code and executed the data analysis pipeline. S.B. wrote the manuscript with contributions from T.S., C.G.B., M.S.G., L.J.V., and M.B.M. The project was supervised by A.R., C.G.B., L.J.V., and M.B.M. Funding was secured by B.G., L.J.V., and M.B.M. All authors assisted with manuscript revision and approved the final submitted version. L.J.V. and M.B.M. contributed equally.

## Competing interests

The authors declare no competing interests.
