## [Transparent Peer Review file · Communications Psychology]

No Disconnection Syndrome after Near-Complete Callosotomy

Corresponding Author: Ms Selin Bekir

Version 0:

Decision Letter:

Dear Ms Bekir,

Thank you for your patience during the peer-review process. Your manuscript titled "No Disconnection Syndrome after Near-Complete Callosotomy" has now been seen by 3 reviewers, and I include their comments at the end of this message. They find your work of interest but raised some important points. We are interested in the possibility of publishing your study in Communications Psychology, but would like to consider your responses to these concerns and assess a revised manuscript before we make a final decision on publication.

We therefore invite you to revise and resubmit your manuscript, along with a point-by-point response to the reviewers. Please highlight all changes in the manuscript text file.

Editorially, we consider it important that the revised manuscript include additional neuroimaging and clinical patient data to address the reviewers' concerns. Please use caution in the interpretation of the findings, staying close to the data and acknowledging limitations.

I am attaching an Editorial Requests Table that details critical reporting requirements for the revised manuscript. Please attend to each item and ensure your manuscript is fully compliant. If your revised manuscript is not aligned with these requests on major issues, such as those concerning statistics, it may be returned to you for further revisions without re-review.

Please submit the following items:

- Revised manuscript
- Point-by-point response to the referees' comments
- Cover letter (as a separate document)
- <https://www.nature.com/documents/nr-reporting-summary.pdf> Nature Research Reporting Summary
- Completed Editorial Request Table (attached).

via this link: Link Redacted .

Additional guidance is available in our style and formatting guide Communications Psychology formatting guide.

Best regards,

Jennifer Bellingtier

Jennifer Bellingtier, PhD
Senior Editor
Communications Psychology

REVIEWER EXPERTISE:

Reviewer #1 neuroplasticity, brain mapping
Reviewer #2 neuropsychology, neuroscience
Reviewer #3 neuropsychology, brain injury

REVIEWER REPORTS:

Reviewer #1 (Remarks to the Author):

This study investigates the neuropsychological consequences of partial versus complete callosotomy in four neurosurgical adult patients, each assessed with lateralized behavioral tasks several years after the surgical procedure. The authors report that one patient with a small remnant of the posterior corpus callosum (splenium fibers) demonstrated preserved performance across the five behavioral tasks and thus showed no signs of disconnection syndrome. In contrast, the three patients who underwent complete callosotomy displayed behavioral deficits consistent with classic interhemispheric disconnection. The authors interpret these findings as evidence that even a minimal splenial remnant may be sufficient to support large-scale interhemispheric functional integration, potentially explaining functional compensation in the patient of interest.

While provocative, the conclusions are based on a single case, lack pre- and early post-operative behavioral and neuroimaging data (diffusion and functional MRI), and rely on bedside rather than controlled experimental paradigms. Nevertheless, the study raises interesting questions about the mechanisms of neuroplasticity that may preserve interhemispheric integration and communication in the context of partial structural disconnection of CC.

Having said that, the methods are clearly described and presented in sufficient detail to allow replication. The statistical approach appears appropriate and aligned with the study's aims. In my opinion, the results are valid and informative; However, the conclusions may be slightly overstated given the reliance on a neuropsychological assessment without complementary MRI data.

Please find below my main comments, which I hope the authors will find constructive in revising the draft.

#1 While the absence of deficits across the five behavioral tasks is notable, and indeed somewhat surprising, it may not be sufficient to conclude that the patient is free of disconnection syndromes. Such syndromes can manifest in both task-dependent and modality-specific ways and thus requires a broader range of neuropsychological assessments to be adequately detected. Furthermore, the tasks used were bedside in nature and thus lacked the experimental precision of controlled paradigms such as tachistoscopic presentations. Having said that, I do not question the relevance of the tasks used or the validity of the reported results, but rather whether the assessment is sufficiently comprehensive and sensitive to support the claim that the patient has fully recovered from the surgical procedure in terms of disconnection syndrome.

#2 An important limitation is the absence of preoperative, and perhaps more crucially, early postoperative, neuropsychological assessments. Without such data, it is not possible to determine whether disconnection syndromes were initially present in the patient of interest and subsequently resolved, which would have provided direct evidence of functional

compensation over time.

#3 It is unfortunate that diffusion MRI data are not available, at least for the patient of interest, as such imaging modality could have helped identify the cortical terminations of the preserved splenial fibers and thus provided more detailed anatomical context for interpreting the lack of expected disconnection syndromes.

#4 In the same vein, resting-state fMRI would have allowed the authors to determine whether homotopic areas, densely interconnected via the corpus callosum, remained functionally synchronized despite callosal damage. In other words, it would have been particularly informative to show whether the three patients with complete callosotomy exhibited disrupted interhemispheric functional connectivity, in contrast to the partially preserved patient. Such evidence would have strengthened the claim that even a small splenial remnant can sustain large-scale interhemispheric integration.

#5 In my view, the authors' general conclusion may be somewhat overstated. Can we truly assert that this study "shows the remarkable functional capabilities of a small, intact posterior callosal remnant years after surgery"? While BT is indeed an interesting case, this observation alone does not definitively demonstrate the splenium's unique role in "reorganized interhemispheric integration". It may be so, but it is equally plausible that the patient is neurophenotypically atypical, or that functional redistribution occurred through alternative pathways (via subcortical nuclei, anterior or posterior commissure, for example) rather than the preserved portion of the splenial fibers. Again, I do not question the interest or value of the case, but it remains difficult to draw strong conclusions about mechanisms of neuroplastic reorganization in the absence of additional MRI-based methods.

Reviewer #2 (Remarks to the Author):

I thank the editor for the opportunity to review this manuscript, which investigates the behavioural and functional consequences of corpus callosotomy in four patients. The study aims to unveil the isolated contribution of the splenium reorganisation in intra- and inter-hemispheric processing across different domains and modalities. The authors challenge classic models of callosotomy functional consequences by showing a surprising degree of spared inter-hemispheric abilities in a patient with partial callosotomy, suggesting the functional reorganisation of the inter-hemispheric pathway and compensatory mechanisms. Nevertheless, some concerns need to be addressed.

Major comments

- The one-to-one association between splenium projections and visual areas has been challenged (e.g. see Friedrich et al <https://doi.org/10.1016/j.neuroimage.2020.117317>), and projections of the splenium could also reach the temporal cortices. This might partially explain the spared inter-hemispheric abilities extending beyond visual functions, compensating, for instance, for the transfer of semantic information.
- The manuscript presents tractographic reconstructions but lacks sufficient methodological detail and visual evidence. It is crucial to provide more slices from the tractography reconstruction and a clearer side-by-side comparison between patients. In particular, it remains unclear whether the poorer sensorimotor performance in patient TJ could be due to damage to tracts outside the splenium. Without clearer anatomical coverage and quantification, interpretations of behavioural dissociations remain speculative. Moreover, no details are provided on the DWI acquisition parameters or tractography reconstruction. Given that the extent of structural disconnection is central to the manuscript's conclusions, this is a major omission and should be addressed explicitly.
- The so-called "speech comprehension" task does not isolate comprehension per se, as it also relies on gesture production, introducing a confound with praxis. The proposed control task, meaningful gesture imitation, does not adequately control for this, as it bypasses meaningless motor planning and execution. A more appropriate control would have involved meaningless gesture imitation or production, which would allow for a clearer dissociation between semantic and motor components. Moreover, the authors' claim that "her right hemisphere may be limited in its ability to comprehend speech" (lines 235-236) is problematic, as the observed deficit might instead reflect a lack of semantic access to gesture meaning, not impaired comprehension. A lateralised auditory comprehension task would have provided a more robust control.
- More generally, the study does not provide a meaningful assessment of language function, since the speech production task also largely replicates the demands of the previously presented visual and sensory processing tasks, without introducing a distinct linguistic component.
- The interpretation of preserved function in patient NR (e.g. bilateral visuospatial processing and speech comprehension) is presented as evidence for rehabilitation-induced plasticity and premorbid bilateral organisation of these functions, while similar preserved performance in patient BT is evidence of splenium reorganisation. If plasticity and premorbid variability are invoked for NR, why not for BT? This asymmetry weakens the theoretical argument. Moreover, BT was retested years after the surgery, and NR seemed to be retested within the year. This big difference could imply a potential recovery also in NR in a few years, raising doubts on the exclusive reorganisation role of the splenium and suggesting the presence of intra-hemispheric reorganisation. Yet this critical variable is not sufficiently considered. The potential contribution of rehabilitation, recovery time, or pre-morbid variability in cognitive profile should be more evenly addressed with a more detailed comparison between the two patients.

- In Figure 3, visuospatial and speech comprehension are missing. Rather than relying on colour-coded performance summaries, the authors should present direct statistical comparisons between patients. Moreover, intra-hemispheric conditions often rely on a single hemisphere per domain, limiting generalisability and interoperability.

- While the behavioural battery is rich in lateralised tasks, the general neuropsychological profile of the patients is incompletely described. A broader cognitive assessment that includes executive functions, memory, and non-lateralised praxis would provide critical context, especially given the individual variability in collosotomy extent and recovery.

Minor comments

- The Introduction (lines 76–81) includes a repetition that should be removed.

- The distinction between intra- and inter-hemispheric conditions depends variably on the input modality and the response modality, creating interpretative ambiguity. A clearer operational definition, perhaps supported by a table, would improve the clarity.

- Some tasks, such as the topognosis task, are insufficiently explained in the main text.

- I recommend reporting time from surgery in months rather than “age at testing”.

Reviewer #3 (Remarks to the Author):

The study aims to explore the brain's capacity to maintain behavioural integration through preserved pathways after surgical interruption of callosal pathways. The authors presented the post-surgical clinical assessment in four patients, three with complete callosotomy and one whose splenium part of the corpus callosum was left intact. This patient presented unexpected behavioural outcomes since the first clinical evaluation. The authors presented a proper methodology for evaluating intra- and interhemispheric connections among visual, tactile, language and visuospatial domains. The aim of the study is relevant and novel. The authors introduce a standardised psychometric profile metric called “Disconnection Score” that could guarantee the comparison of disconnection profiles among single case studies. The methodology is solid and clearly explained. The results comparing commonalities and differences, focusing on the left intact splenium, shed new light on the role of preserved pathways reflected in preserved task abilities. This could lead to further structural connectivity studies for diagnostic and future rehabilitative approaches after surgery.

Overall, the work has been conducted with the proper methodology considering the small group and the author's aims. The manuscript is valid, for the importance of the topic, and for encouraging group studies on syndromes like the ones that characterise alterations in brain functional connections.

Below are a few theoretical comments and some suggestions that may be considered to improve the work.

1. Authors reported in methods that the anterior, posterior and hippocampal commissures remained intact in all patients. In the Introduction section, I suggest mentioning the role of the commissural auxiliary pathways when the corpus callosum is absent or altered in its anatomy or function. Commissural auxiliary pathway could support residual interhemispheric integration of visuospatial and sensorimotor functions (i.e. spatial orientation, visual processing, and motor coordination). Still, they cannot fully compensate for the absence of the corpus callosum. This provides relevant insight, considering that your results highlight that the preservation of the splenium appears to be the only difference among patients that may allow for inferences about its role in supporting reorganisation processes critical for interhemispheric functioning.

2. The Methods and discussion sections are well-crafted, clearly and concisely written, with great detail and care. The entire manuscript is generally easy to follow for researchers and readers from a clinical background. A strong understanding of clinical assessment and evidence is undoubtedly necessary, yet the whole topic of disconnection syndrome remains complicated, and I appreciate how the authors have presented the entire topic. However, I suggest aligning the writing style of the Introduction and Results sections with that of the Method and Discussion, as this would elevate the overall quality of the manuscript. For example, lines 85-89 could be presented in a more effective way that would better highlight the originality of the study.

3. Table 1: Information regarding the participants' education level may be relevant. Furthermore, I suggest including details on the participants' sensory and motor integrity. While these aspects are likely reflected in the performance of the intrahemispheric tasks, explicitly reporting them would provide a more complete and informative clinical profile.

4. In the Results section and Table 2, the number of trials is reported with differences across patients and between intra- and interhemispheric tasks. In the Table 2 caption, the authors mention that this limitation regarding unmatched trial numbers would be discussed in the Results and Discussion sections. While it is possible to infer what they are referring to, and I understand the challenges involved in clinical assessments with patients, it is necessary to explain clearly why the number of trials differs among patients and between intra- and interhemispheric tasks.

5. Line 173-183: The authors' interpretation of TJ's perceptions appears reasonable and is likely to be broadly agreed upon. However, they might consider that her behaviour could appear unusual to readers who can only access limited information about her clinical history. Furthermore, Table 2, which reports her performance, raises several important considerations. In the task, the patient was asked to name everyday objects placed in her left hand, and she reportedly claimed to feel nothing

in that hand. Is this accurate? Could this explain why TJ completed more trials than the other participants in the Topognosis task? I suggest including a few lines in the Discussion section addressing TJ's behaviour. Additionally, it would be interesting to know how TJ performed in the control-mimicking task concerning her frustration and motivational issues.

6. Lines 208-213. The interpretation provided by the authors regarding NR's performance is compelling and could be widely accepted. However, the authors should consider whether any clinical information is available regarding the patient's performance on that specific task within the six months following surgery. For instance, if the patient performed the task shortly after surgery with a low success rate but achieved 100% success at your six-month evaluation, it might be appropriate to discuss the possibility of a learning effect. Conversely, if the patient already performed well from the outset, this would support the bilateral representation for the visuospatial processing hypothesis. The successful performance guided by "Regular rehabilitation exercises involving similar tasks" reflects a generalisation of the task, an excellent outcome 6 months after surgery. The patient is young and his brain plasticity certainly played a role but – to entertain a conversation with the authors - Might this explanation be considered somewhat weak, given the specificity of the WAIS Block Design Test according to your experience?

If previous data on the Block Design Test or other neuropsychological assessments—such as memory tasks—were available, they would provide substantial added value to the interpretation of the findings.

7. Bayne 2008 doesn't seem to be mentioned in the text but only reported in the reference list. I suggest checking and revising citations' sequential numbering to meet the journal's standards before reaching the final stages.

8. Figure 4 and the discussion section deserve a dedicated appreciation: In these branches of clinical research, current and future studies can only benefit from past single-case reports, as they provide valuable points of comparison and help strengthen the evidence base on the topic.

If you experience problems in linking your ORCID, please contact the Platform Support Helpdesk.

Version 1:

Decision Letter:

Dear Ms Bekir,

Your manuscript titled "No Disconnection Syndrome after Near-Complete Callosotomy" has now been seen by our reviewers, whose comments appear below. In light of their advice I am delighted to say that we are happy, in principle, to publish a suitably revised version in Communications Psychology.

We therefore invite you to revise your paper one last time to address the remaining concerns of our reviewers and a list of editorial requests. At the same time we ask that you edit your manuscript to comply with our format requirements and to maximise the accessibility and therefore the impact of your work.

EDITORIAL REQUESTS:

SUBMISSION INFORMATION:

OPEN ACCESS:

* DATA AVAILABILITY:

Link Redacted

Best regards,

Jennifer Bellingtier

Jennifer Bellingtier, PhD
Senior Editor
Communications Psychology

REVIEWER EXPERTISE:

Reviewer #1 neuroplasticity, brain mapping
Reviewer #2 neuropsychology, neuroscience
Reviewer #3 neuropsychology, brain injury

REVIEWERS' COMMENTS:

Reviewer #1 (Remarks to the Author):

Dear authors,

I have carefully read the revised manuscript. Thank you very much for your efforts in addressing my concerns. In my opinion, the manuscript is now more nuanced, and the limitations clearly acknowledged. I have no further comments. Congratulations on the excellent work.

Reviewer #2 (Remarks to the Author):

I would like to thank the authors for the clarifications provided. The manuscript has substantially improved. The addition of the neuropsychological summary table (which could be included as supplementary material) and the new figure are both welcome and help clarify the results.

I only have one remaining minor comment on the methods of the DWI acquisition.

Although the tract image is presented primarily for visualisation purposes, the manuscript still needs to include the essential diffusion acquisition details and a brief description of how the tracts were obtained. If the authors prefer not to expand the main text, this could be briefly stated in the Methods section with reference to supplementary material for full details. Showing the tractography without reporting how these tracts were derived leaves a methodological gap that should be clarified.

Once this point is addressed, I believe the manuscript will be suitable for publication.

Reviewer #3 (Remarks to the Author):

I approve the authors' responses and the competence with which they have provided them. I appreciated the authors' clinical experience in addressing the issues raised by the reviewers, as well as in providing specific answers in addition to the direct changes in the manuscript. This work meets the journal's standards and contributes to promoting future research and discussion on how the brain may adapt to perturbation through residual pathways.

Open Access This Peer Review File is licensed under a Creative Commons Attribution 4.0 International License, which permits use, sharing, adaptation, distribution and reproduction in any medium or format, as long as you give appropriate credit to the original author(s) and the source, provide a link to the Creative Commons license, and indicate if changes were

made.

Replies to all Referees

Overview

We thank all 3 Referees for their enthusiasm for our work, promising comments and helpful suggestions to improve the paper. Below, we addressed each point raised by the referees.

Changes to the manuscript are highlighted in blue in the revised version of the manuscript.

*For easier tracking, we highlighted **previously existing sections in light blue** and **newly added ones in dark blue** whenever we quote them in this response-to-referees document*

We'd like to note that the figures have been renumbered due to manuscript reorganization and addition of a new Figure. All responses below follow the new numbering, and for clarity we indicate the original numbering in parentheses (e.g., 'Figure 2 (previously Figure 1)').

Reviewer 1 comments

1. While the absence of deficits across the five behavioral tasks is notable, and indeed somewhat surprising, it may not be sufficient to conclude that the patient is free of disconnection syndromes. Such syndromes can manifest in both task-dependent and modality-specific ways and thus requires a broader range of neuropsychological assessments to be adequately detected. Furthermore, the tasks used were bedside in nature and thus lacked the experimental precision of controlled paradigms such as tachistoscopic presentations. Having said that, I do not question the relevance of the tasks used or the validity of the reported results, but rather whether the assessment is sufficiently comprehensive and sensitive to support the claim that the patient has fully recovered from the surgical procedure in terms of disconnection syndrome.

The referee here addresses the limitations of bedside testing procedures, which offer less controlled experimental conditions, e.g., compared to paradigms using lateralized visual tachistoscopic presentation of stimuli. We fully agree with the referee, that the bedside testing procedures employed here may be more prone to miss residual disconnection effects.

We now address and discuss this point in our revised discussion section (lines 535-538). With this section, we hope to clarify that we cannot definitively rule out the absence of any disconnection syndrome. It is possible that our bedside tasks were not sensitive enough to detect subtle deficits arising from anterior CC loss. As the Referee suggests, future work with more rigorous, targeted paradigms will be important for probing the anterior CC's contribution to cognition in this case.

*“The direction of the callosal resection—**anterior-to-posterior versus posterior-to-anterior**—might also be an important factor.⁴² One of the proposed*

evolutionary origins of the CC is argued to be midline fusion for the sensory cortices,¹ with humans heavily relying on sensorimotor processes to interact with the world. Thus, posterior inter-hemispheric integration might better preserve the system's integrity and may allow for more effective compensation than vice versa. What, then, is the role of the anterior CC? Our tasks may have been insensitive to effectively probe this question. While we cannot definitively rule out the absence of *any* disconnection syndromes in BT*, he showed no such signs on our battery of bedside assessments. Moving forward, more rigorous and targeted testing with BT* is needed to better probe the anterior CC's role in cognition. “

2. An important limitation is the absence of preoperative, and perhaps more crucially, early postoperative, neuropsychological assessments. Without such data, it is not possible to determine whether disconnection syndromes were initially present in the patient of interest and subsequently resolved, which would have provided direct evidence of functional compensation over time.

The referee is concerned about the absence of pre- and early postoperative neuropsychological assessments, which limits the ability to determine whether disconnection syndromes were initially present and later resolved or were not present even in the immediate post-operative phase. We now address this explicitly by adding the following limitation (lines 576–581):

“the *lack of longitudinal* data prevents us from determining whether BT*'s intact performance reflects an absence of disconnection syndrome all along or their resolution through network reorganization over the years post-surgery. While network reorganization remains the most plausible explanation given the existing reports of selective impairments in other splenium cases¹⁸, we cannot definitively establish BT*'s symptom trajectory.”

Given prior case reports of splenium-only patients (e.g., Risse et al., ref. 18) who were tested shortly after surgery and exhibited selective disconnection syndromes, we think it is likely that BT also experienced such symptoms initially, which then resolved over time. However, this inference is based on previous similar case reports and cannot be empirically confirmed in our case.

We also revised our conclusion paragraph to further emphasize this (lines 617-625):

“In conclusion, our study suggests that even a small, intact posterior callosal remnant can sustain remarkable functional capabilities years after surgery. While BT* is a single case, and the absence of longitudinal data prevents us from

determining whether this integration was preserved immediately post-surgery or emerged through network reorganization over time, this observation nevertheless highlights the splenium's remarkable role in such integration. Our current findings support the notion that inter-hemispheric structure-function relationships in the human brain do not follow a linear pattern where behavioral disruptions scale with the extent of structural (callosal) resection. Instead, minimal preserved structural connectivity may maintain broad integration beyond what our classical anatomical models predict.”

3. It is unfortunate that diffusion MRI data are not available, as least for the patient of interest, as such imaging modality could have helped identify the cortical terminations of the preserved splenial fibers and thus provided more detailed anatomical context for interpreting the lack of expected disconnection syndromes.

We fully agree that diffusion MRI would provide valuable additional anatomical context, particularly for identifying the cortical terminations of the preserved splenial fibers in this patient. Unfortunately, robust identification of inter-hemispheric fibers is a nontrivial task even with sophisticated multi-shell dMRI sequences, and sufficiently high-resolution diffusion data were not available for this patient: we were only able to collect DTI data with 30 diffusion-encoding gradients, which precludes the ability to resolve inter-hemispheric fibers that may turn at severe angles and pass through junctions of crossing fibers in association cortex. Thus, any inter-hemispheric tractography analyses based on low resolution data in a single subject would be highly dependent on tracking parameters and therefore inconclusive, i.e., may yield many false positives (e.g., Jones, Knösche & Turner, 2013, NeuroImage 73:239-254: <https://doi.org/10.1016/j.neuroimage.2012.06.081>). To address the Referee’s point that such imaging data would be an invaluable addition, we have added the following part to the limitations section (lines 590-596):

“While the splenium is traditionally associated with visual information integration, recent evidence suggests its fibers also project to temporal cortices.⁴⁶ It is possible that what is transferred may not be the sensory input, but more abstract, conceptual representations. More rigorous behavioral testing, alongside event-related functional neuroimaging and high-resolution diffusion MRI—including tractography analyses to identify cortical termination points—will be necessary to further delineate these mechanisms and clarify the functional contributions of preserved splenial pathways.”

We want to emphasize that diffusion tractography images included in Figure 2 (previously Figure 1) were presented for visualization purposes only and are unfortunately insufficient for further formal quantitative analysis (a similar point was

raised by Reviewer 2, Comment #2 so, please also see our response there). As we mentioned above, inter-hemispheric tractography to identify cortical endpoints is nontrivial—especially in a single patient and without sufficiently high-resolution diffusion data, which would be highly parameter-dependent (Calabrese et al., 2014 <https://pubmed.ncbi.nlm.nih.gov/25044786/>). To do this analysis justice, higher-resolution diffusion MRI would be required, which we plan to acquire in the future.

4. In the same vein, resting-state fMRI would have allowed the authors to determine whether homotopic areas, densely interconnected via the corpus callosum, remained functionally synchronized despite callosal damage. In other words, it would have been particularly informative to show whether the three patients with complete callosotomy exhibited disrupted interhemispheric functional connectivity, in contrast to the partially preserved patient. Such evidence would have strengthened the claim that even a small splenial remnant can sustain large-scale interhemispheric integration.

This is an important consideration and we indeed addressed these questions in a separate resting-state fMRI study (Santander et al., ref. 40, in-press at the Proceedings of the National Academy of Sciences: <https://www.biorxiv.org/content/10.1101/2025.02.14.638327v3>). In brief, the partial patient (BT), who retains a small splenial remnant, showed largely preserved inter-hemispheric functional connectivity across multiple networks and metrics of inter-hemispheric FC (relative to healthy controls from the Human Connectome Project). In contrast, the complete-callosotomy patients exhibited marked disruptions. These observations further support our claim that even a small posterior callosal remnant can sustain large-scale inter-hemispheric coupling, consistent with BT's intact behavioral profile. We have now added the following paragraph to the Discussion (lines 515-521)*

“While subcortical structures have also been proposed as alternative routes for such reorganization,⁴¹ all patients in our cohort featured intact subcortical pathways. Yet only BT* showed functional unity across all tested domains. This suggests that it is the presence of the splenium—rather than intact subcortical pathways alone—may enable functional unity across hemispheres.

Further support for this notion stems from resting-state functional connectivity analyses which revealed that BT*'s inter-hemispheric connectivity was largely sustained.⁴⁰ Various measures of functional network integration resembled that of healthy controls with an intact corpus callosum, in stark contrast to complete callosotomy patients (see Santander et al.⁴⁰ for further details). These observations show that even small posterior callosal remnants can sustain widespread inter-hemispheric functional coupling, consistent with his intact behavioral performance.”

5. In my view, the authors' general conclusion may be somewhat overstated. Can we truly assert that this study “shows the remarkable functional capabilities of a small, intact posterior callosal remnant years after surgery”? While BT is indeed an interesting case, this observation alone does not definitively demonstrate the splenium’s unique role in “reorganized interhemispheric integration”. It may be so, but it is equally plausible that the patient is neurophenotypically atypical, or that functional redistribution occurred through alternative pathways (via subcortical nuclei, anterior or posterior commissure, for example) rather than the preserved portion of the splenial fibers. Again, I do not question the interest or value of the case, but it remains difficult to draw strong conclusions about mechanisms of neuroplastic reorganization in the absence of additional MRI-based methods.

We appreciate this thoughtful comment. We agree with the Referee’s concerns, and we now accordingly addressed each of the specific concerns:

- It may be so, but it is equally plausible that the patient is neurophenotypically atypical.

It is possible that BT is neurophenotypically atypical. This relates to the issue of individual differences and the single-case nature of our study. We explicitly acknowledge this limitation in our manuscript (lines 609-610):

“...our findings are limited in *generalizability* due to the single-subject nature of BT*’s case.”

- or that functional redistribution occurred through alternative pathways (via subcortical nuclei, anterior or posterior commissure, for example) rather than the preserved portion of the splenial fibers.

*Alternatively, inter-hemispheric transfer could have been mediated through alternative pathways, such as subcortical nuclei or other commissures. Then again, these pathways were intact in **all patients** reported here. Yet **only BT** showed preserved inter-hemispheric behavioral performance across all domains. This makes the splenium the most parsimonious explanation. Here is the section we address this, in the manuscript (lines 510-514):*

“While subcortical structures have also been proposed as alternative routes for such reorganization,⁴¹ all patients in our cohort featured intact subcortical pathways. Yet only BT* showed functional unity across all tested domains. This suggests that it is

the presence of the splenium—rather than intact subcortical pathways alone—may enable functional unity across hemispheres”

- ...it remains difficult to draw strong conclusions about mechanisms of neuroplastic reorganization in the absence of additional MRI-based methods.

We fully agree that additional neuroimaging approaches would be valuable to better characterize the mechanisms supporting BT’s intact behavioral performance. To address this, we have added a brief summary of our resting-state fMRI findings (see our response to Comment #4 above) and added the following section to our limitations (lines 587-596):

“while BT* showed no disconnection syndrome on our battery of bedside assessments, our current methodology does not allow us to pinpoint the underlying mechanisms for this functional unity. In other words, the precise nature of what is being integrated remains unclear. While the splenium is traditionally associated with visual information integration, recent evidence suggests its fibers also project to temporal cortices.⁴⁶ It is possible that what is transferred may not be the sensory input, but more abstract, conceptual representations. More rigorous behavioral testing, alongside event-related functional neuroimaging and high-resolution diffusion MRI—including tractography analyses to identify cortical termination points—will be necessary to further delineate these mechanisms and clarify the functional contributions of preserved splenial pathways.”

- Concern with the overstatement:

In light of this comment, we have tempered the language of our conclusions to avoid any statements which may facilitate overinterpretation of our results. Specifically, we replaced terms such as “shows” or “demonstrates” with “suggests” (lines 617-625):

“In conclusion, our study suggests that even a small, intact posterior callosal remnant can sustain remarkable functional capabilities years after surgery. While BT* is a single case, and the absence of longitudinal data prevents us from determining whether this integration was preserved immediately post-surgery or emerged through network reorganization over time, this observation nevertheless highlights the splenium's remarkable role in such integration. Our current findings support the notion that inter-hemispheric structure-function relationships in the human brain do not follow a linear pattern where behavioral disruptions scale with the extent of structural (callosal) resection. Instead, minimal preserved structural connectivity may maintain broad integration beyond what our classical anatomical models predict.”

Taken together, we hope that the revised manuscript presents a more neutral interpretation, explicitly acknowledges alternative mechanisms and study limitations, and clarifies that our findings suggest that a small posterior callosal remnant may support inter-hemispheric integration years after surgery. Nevertheless, BT's unexpectedly intact behavioral outcome underscores the potential value of this unique case in opening new avenues for probing the mechanisms of inter-hemispheric integration.

Reviewer 2 comments

1. The one-to-one association between splenium projections and visual areas has been challenged (e.g. see Friedrich et al <https://doi.org/10.1016/j.neuroimage.2020.117317>), and projections of the splenium could also reach the temporal cortices. This might partially explain the spared inter-hemispheric abilities extending beyond visual functions, compensating, for instance, for the transfer of semantic information.

This is an excellent point, and we thank the Referee for raising it and for sharing the citation. We have revised our limitation section to add the following section (lines 587-596):

“while BT showed no disconnection syndrome on our battery of bedside assessments, our current methodology does not allow us to pinpoint the underlying mechanisms for this functional unity. In other words, the precise nature of what is being integrated remains unclear. While the splenium is traditionally associated with visual information integration, recent evidence suggests its fibers also project to temporal cortices.⁴⁶ It is possible that what is transferred may not be the sensory input, but more abstract, conceptual representations. More rigorous behavioral testing, alongside event-related functional neuroimaging and high-resolution diffusion MRI—including tractography analyses to identify cortical termination points—will be necessary to further delineate these mechanisms and clarify the functional contributions of preserved splenial pathways.”*

2. The manuscript presents tractographic reconstructions but lacks sufficient methodological detail and visual evidence. It is crucial to provide more slices from the tractography reconstruction and a clearer side-by-side comparison between patients. In particular, it remains unclear whether the poorer sensorimotor performance in patient TJ could be due to damage to tracts outside the splenium. Without clearer anatomical coverage and quantification, interpretations of behavioural dissociations remain speculative. Moreover, no details are provided on the DWI acquisition parameters or tractography reconstruction. Given that the extent of structural disconnection is central to the manuscript's conclusions, this is a major omission and should be addressed explicitly.

We appreciate this important concern.

To clarify, the tractography images in Figure 2 (previously Figure 1) were included solely for illustrative purposes, not for any formal quantitative analysis. Our goal was to provide a visual depiction of midline anatomy—specifically, to highlight the intact anterior commissure in all patients, the residual splenial fibers in BT, and the absence of callosal fibers in the other cases. For this reason, tractography parameters were individually adjusted for each patient to achieve the clearest mid-sagittal visualization— analogous to selecting different T1-weighted planes for optimal visualization in individual brains. We have now made this illustrative intent explicit in the Figure caption (lines 149-156):*

“Figure 2 (previously Figure 1) Anatomical Scans of the Patients. Midsagittal T1-weighted MRI images with overlaid DTI-based fiber tractography are shown for each patient. Note that diffusion tractography is shown solely for visualization; only select commissural fibers—the anterior commissure and corpus callosum—as well as some cerebellar and brainstem fibers, are displayed. Tractography parameters were adjusted for each patient to provide the clearest midsagittal depiction. The fiber orientations are color-coded: red for left-right, green for anterior-posterior, and blue for superior-inferior directions. All patients exhibit an intact anterior commissure (white arrow), but only patient BT* retains a preserved splenium (yellow arrow), in contrast to the complete sectioning of the corpus callosum in the remaining three patients.“

We fully agree that a formal, quantitative diffusion tractography analysis could yield valuable insights (see also our response to Reviewer 1, Comment #3). Unfortunately, the available diffusion data lack the spatial resolution necessary for robust inter-hemispheric fiber tracking; single-subject analyses in this context would be inconclusive. We have added this limitation to the manuscript (lines 593-596). We hope to acquire such high resolution diffusion MRI with this cohort in the future.

“More rigorous behavioral testing, alongside event-related functional neuroimaging and high-resolution diffusion MRI—including tractography analyses to identify cortical termination points—will be necessary to further delineate these mechanisms and clarify the functional contributions of preserved splenial pathways.

Overall, we think the tractography overlays provided a clear and intuitive visualization of each patient’s callosotomy profile. However, we recognize that their inclusion in Figure 2 (previously Figure 1) may lead to confusion. If they are proving more distracting than informative, we are happy to omit them and present only the T1-weighted images in Figure 2 (previously Figure 1).

As for TJ: (in response to “it remains unclear whether the poorer sensorimotor performance in patient TJ could be due to damage to tracts outside the splenium” comment by the Referee):

*We would not characterize TJ as having poor sensorimotor performance overall. Her deficits emerged **only** when inter-hemispheric transfer was required. For example, in topognosis (touch localization), she was 100% accurate when the hand that received the stimulus also produced the response (i.e., the intra-hemispheric uncrossed condition: for left hand stimulated and left hand responded [LL], or [RR], accuracy was 26/26 combined). By contrast, performance dropped only in the inter-hemispheric (crossed) condition [LR], or [RL]; 4/30). Similarly, her left hand (controlled by the Right Hemisphere) successfully imitated all hand gestures (9/9) with eyes open but was only impaired when eyes were closed and performance depended on verbal instructions. This suggests that the Right Hemisphere (RH) could only respond appropriately to visual instructions but not to verbal ones in this task. In the visuospatial block design, her nondominant left hand excelled (5/6—1 was mirror imaged), whereas her dominant right hand was impaired this time.*

In sum, TJ performed at ceiling when: (i) input and output remained within the same hemisphere irrespective of which hand she was using, and (ii) when hand performance aligned with hemispheric specialization (e.g., left hand excelled in visuospatial tasks—RH dominance; right hand excelled in speech comprehension—LH dominance).

Lastly, although we did not carry out a quantitative diffusion MRI analysis, we conducted a separate resting-state functional connectivity analysis in this cohort (see our response to Reviewer 1, Comment #4). We now reference this study and its findings more explicitly in this manuscript, to support our conclusion that even a small posterior callosal remnant can support large-scale inter-hemispheric coupling, consistent with BT’s preserved behavioral profile.*

3. The so-called “speech comprehension” task does not isolate comprehension per se, as it also relies on gesture production, introducing a confound with praxis. The proposed control task, meaningful gesture imitation, does not adequately control for this, as it bypasses meaningless motor planning and execution. A more appropriate control would have involved meaningless gesture imitation or production, which would allow for a clearer dissociation between semantic and motor components. Moreover, the authors’ claim that “her right hemisphere may be limited in its ability to comprehend speech” (lines 235-236) is problematic, as the observed deficit might instead reflect a lack of

semantic access to gesture meaning, not impaired comprehension. A lateralised auditory comprehension task would have provided a more robust control.

We thank the Referee for this very insightful comment. Our 'hand gesture' task includes both meaningless gestures (e.g., 'three fingers to the side,' 'four fingers up') and meaningful ones (e.g., 'thumbs up,' 'thumbs down'). Nevertheless, we conceptualize the relevant processing steps for our speech comprehension task as follows:

- (1) Speech perception (auditory input) *
- (2) Semantic comprehension *
- (3) Motor planning/execution*

Our "mimicking" control task ("copy-what-I-do" task) was designed to isolate motor/praxis demands while equating gesture knowledge by using the same gesture set in both tasks. If the patient can perform this control task but fails on the verbal command version, we suggest this effectively rules out step (3) "Motor planning/execution" as the primary locus of impairment.

We agree that more rigorous meaningless gesture imitation could further reduce reliance on pre-existing gesture representations, and thus, increasing trial numbers for those trials would be a valuable addition in future work. Nonetheless, given that the same set of gestures was used in both tasks, the critical contrast we think is between the following:

Control mimicking task: Motor ability + (gesture knowledge)

*Verbal command task: Motor ability + (gesture knowledge) + **speech comprehension** (steps 1 and 2)*

The patient's intact performance in the control task (eyes open) alongside impaired performance on the verbal command task (eyes closed) suggests that the deficit likely lies somewhere in the auditory-to-semantic pathway (steps 1 and/or 2)—but not in 3.

While we agree it is difficult to pinpoint the precise stage of breakdown (phonological processing vs. semantic access), we think that this pattern constitutes evidence of impaired comprehension. We have now clarified this point in the manuscript by adding the following paragraph under limitations (lines 598-606):

"Our "speech comprehension" task likely engaged a multi-step process: (1) auditory speech perception, (2) semantic comprehension, and (3) motor planning/execution. The control gesture imitation task rules out step 3—praxis or motor deficits—if the patient was able to accurately mimic gestures on sight. Thus, failure to perform gestures on verbal command, despite intact imitation, points to a disruption in the auditory-to-semantic pathway (steps 1 and/or 2), which our current design cannot further dissociate. Future studies using refined controls, such as more

rigorous testing of meaningless gesture imitation or lateralized auditory tasks, can help localize where in the process it breaks down (phonological input vs semantics).”

4. More generally, the study does not provide a meaningful assessment of language function, since the speech production task also largely replicates the demands of the previously presented visual and sensory processing tasks, without introducing a distinct linguistic component.

We agree with this concern and think that our speech production tasks could have benefited greatly from including a non-verbal response modality (e.g., left-hand responses during finger perimetry) to ensure that it is the verbalization that is impaired and not visual or sensory processing. For example, in the Finger Perimetry task, if the right hemisphere could correctly show how many fingers it saw using the left hand but failed to name it, this would more directly implicate a speech production impairment rather than a perceptual one.

While extensive previous work showed similar results (Gazzaniga & Sperry, 1962, <https://pubmed.ncbi.nlm.nih.gov/13946939/>; Gazzaniga, 2000, <https://pubmed.ncbi.nlm.nih.gov/10869045/>), this was not systematically implemented across our current cohort, and we therefore acknowledge this as a limitation of the current findings. Nevertheless, intact performance on other nonverbal tasks (e.g., topognosis, gesture imitation) indicates that the right hemisphere was capable of perceiving and responding to stimuli when verbalization was not required.

To address this concern, we included the following section to our limitations (lines 596-610, combined with our response to above comment #3):

“with respect to our language tasks, we make inferences about speech comprehension and production, but more rigorous testing is needed to evaluate these processes in more detail. Our “speech comprehension” task likely engaged a multi-step process: (1) auditory speech perception, (2) semantic comprehension, and (3) motor planning/execution. The control gesture imitation task rules out step 3—praxis or motor deficits—if the patient was able to accurately mimic gestures on sight. Thus, failure to perform gestures on verbal command, despite intact imitation, points to a disruption in the auditory-to-semantic pathway (steps 1 and/or 2), which our current design cannot further dissociate. Future studies using refined controls, such as more rigorous testing of meaningless gesture imitation or lateralized auditory tasks, may help to further delineate why patients were not able to successfully perform the task (e.g., whether it is related to phonological input vs

semantics). Similarly, our speech production tasks would benefit from the addition of a systematic non-verbal response condition (e.g., left-hand responses during finger perimetry) to ensure that observed deficits indeed reflected impaired verbalization rather than failures in visual or sensory processing.

5. The interpretation of preserved function in patient NR (e.g. bilateral visuospatial processing and speech comprehension) is presented as evidence for rehabilitation-induced plasticity and premorbid bilateral organisation of these functions, while similar preserved performance in patient BT is evidence of splenium reorganisation. If plasticity and premorbid variability are invoked for NR, why not for BT? This asymmetry weakens the theoretical argument. Moreover, BT was retested years after the surgery, and NR seemed to be retested within the year. This big difference could imply a potential recovery also in NR in a few years, raising doubts on the exclusive reorganisation role of the splenium and suggesting the presence of intra-hemispheric reorganisation. Yet this critical variable is not sufficiently considered. The potential contribution of rehabilitation, recovery time, or pre-morbid variability in cognitive profile should be more evenly addressed with a more detailed comparison between the two patients.

We thank the Referee for raising this excellent point. We agree that rehabilitation and premorbid bilateral representations are important factors to consider, especially for functions with hemispheric dominance such as speech comprehension and visuospatial processing. We have now added this important aspect to our limitations section (lines 576-586):

“, the lack of longitudinal data prevents us from determining whether BT's intact performance reflects an absence of disconnection syndrome all along or their resolution through network reorganization over the years post-surgery. While network reorganization remains the most plausible explanation given the existing reports of selective impairments in other splenium cases,¹⁸ we cannot definitively establish BT*'s symptom trajectory. Similarly, we cannot fully rule out alternative factors such as rehabilitation-induced plasticity or premorbid bilateral representations. However, such explanations are highly unlikely to account for all task domains. For example, while premorbid bilateral organization may support functions with hemispheric dominance (e.g., speech comprehension or visuospatial processing), they are insufficient to explain preserved performance on tasks such as topognosis that strongly rely on inter-hemispheric transfer. “*

6. In Figure 3, visuospatial and speech comprehension are missing. Rather than relying on colour-coded performance summaries, the authors should present direct statistical

comparisons between patients. Moreover, intra-hemispheric conditions often rely on a single hemisphere per domain, limiting generalisability and interoperability.

Please see below for our detailed responses:

- In Figure 3, visuospatial and speech comprehension are missing

In the manuscript, we note the omission of visuospatial and speech comprehension from the color-coded performance summaries (lines 326-331):

“We only included disconnection scores for tasks with clearly defined chance levels and binomial test outcomes. WISC-R spatial block design and hand gestures tasks were not included here because the combination of limited trial numbers and undefined chance levels significantly reduced our statistical power, making the statistical detection of disconnection syndrome challenging in these tasks, even though qualitative evidence supports their presence (see **Results**).“

- Rather than relying on colour-coded performance summaries, the authors should present direct statistical comparisons between patients

*In this study, we did not perform statistical comparisons **between** patients, as the limited number of trials and very small sample size did not allow for any quantitative between-patient analyses. Instead, we based our analyses on within-patient comparisons of intra- (uncrossed) versus inter- (crossed) conditions for each task. The color-coding was intended to provide an intuitive summary of these statistically validated within-subject contrasts.*

- intra-hemispheric conditions often rely on a single hemisphere per domain, limiting generalisability and interoperability.

Inspired by the referee’s comment, we have now added a new Figure (see response to Comment #9) which clarifies intra- versus inter-hemispheric conditions in each task more clearly. We include both RH-dominant (e.g., WISC-R block design) and LH-dominant tasks, providing a useful contrast—for example, cases where the nondominant left hand outperforms the right hand and vice-versa. Our primary aim was to compare inter-hemispheric (crossed) versus intra-hemispheric (uncrossed) trials, irrespective of modality, in line with conventions in split-brain research (please also see comment #9 for a more detailed discussion). We hope the addition of this new figure makes our definitions more transparent and helps to reduce interpretative ambiguity for our future readers.

7. While the behavioural battery is rich in lateralised tasks, the general neuropsychological profile of the patients is incompletely described. A broader cognitive assessment that

includes executive functions, memory, and non-lateralised praxis would provide critical context, especially given the individual variability in collosotomy extent and recovery.

We thank the referee for raising this point and totally agree that further neuropsychological information may benefit the interpretation of our current findings. Therefore, we now report three standard tasks to provide context for patients' general cognitive profile. WAIS-IV Digit Span Forward (Working Memory) evaluates short-term auditory-verbal memory by asking participants to repeat sequences of digits in the same order. WAIS-IV Digit Span Backward (Executive Function) measures working memory and executive control by requiring participants to repeat sequences of digits in reverse order. For both subsets, the raw score reflects the number of correctly repeated items; the span length (Span) indicates the longest sequence recalled correctly. Percentile ranks (Pr) are reported relative to age-normative data alongside descriptive categories (Below Average <16th, Low Average 16-31st, Average 31-69th, High Average 69-84th, Above Average >84th). Apraxia Screen of Tulia (AST; Praxis, right hand only) assesses praxis through pantomime of both communicative and tool-use gestures (e.g., waving, brushing teeth, combing hair, using a screwdriver). Correct trials out of total number of trials are shown.

Domain	Task	description	BT*	NR	TJ	LJ
Working Memory	WAIS-IV Digit Span	Forward	Raw: 5 Span: 4 Pr1 Below average	Raw: 5 Span: 4 Pr2 Below average	Raw: 8 Span: 6 Pr38 Average	Raw: 8 Span: 5 Pr18 Low average
Executive Function	WAIS-IV Digit Span	Backward	Raw: 6 Span: 3 Pr7 Below average	Raw: 5 Span: 3 Pr4 Below average	Raw: 6 Span: 3 P47 Average	Raw: 3 Span: 2 Pr2 Below Average
Praxis	Apraxia Screen of Tulia (AST)	Pantomime, right-hand	5/5	5/5	5/6	5/6

We added the following part to our Methods section (lines 134-138):

“Post-operative general neuropsychological testing indicated performance ranging from below average to average on working memory and executive function tasks (WAIS-IV Digit Span, forward and backward),²¹ with BT and NR performing in the below-average range and TJ and LJ falling in the low-average to average range. Dominant-hand praxis was intact in all patients, as assessed by the Apraxia Screen of Tulia (AST).²²”

For clarity and readability, we did not include this new table in the main manuscript. Instead, we hope that the above summary sentence provides the necessary context.

However, if the Referee is convinced that inclusion of this table would be particularly helpful to our future readers, we are happy to add it to our revised manuscript.

In addition, we added the following paragraph to our Results section (lines 338-342) to highlight that the patients' general sensory and motor integrity was preserved, as evidenced by performance in the intra-hemispheric (uncrossed) trials. The contrast with ceiling-level performance in these uncrossed conditions indicates that the disruptions observed in inter-hemispheric (crossed) trials are likely reflect the demands of inter-hemispheric integration, rather than a broad neuropsychological impairment.

“Across all intra-hemispheric (uncrossed) conditions, patients performed significantly above chance (overall $p < 0.05$; individual p -values per task are reported below). These results indicate intact sensory and motor integrity when tasks did not require inter-hemispheric information integration, whereas performance typically broke down once such integration was necessary.”

8. The Introduction (lines 76–81) includes a repetition that should be removed.

We thank the Referee for this suggestion. We have removed the repetition and revised the relevant sentences in the introduction as follows (lines 74-85):

*“The CC is thought to be topographically organized, where anterior callosal fibers project to anterior cortical regions like the prefrontal cortex, and posterior fibers project to posterior regions like the occipital lobe.⁶ As a result, the CC is argued to have a modality-specific organization with different subsections—from anterior to posterior: rostrum, genu, body, isthmus, and splenium—are hypothesized to facilitate information integration in distinct cognitive and sensorimotor domains.^{7,8} The splenium, the most posterior section of the CC, is typically implicated in visual integration, while the posterior midbody is thought to be involved in tactile integration.⁹⁻¹¹ Evidence from partial callosotomy cases sparing these posterior subregions supports this notion.^{7,12-15} However, in most previous partial callosotomy patients, both the splenium and posterior midbody remain intact, rendering a precise functional distinction impossible. The question thus arises: which kinds of information can the splenium *alone* integrate across hemispheres?”*

9. The distinction between intra- and inter-hemispheric conditions depends variably on the input modality and the response modality, creating interpretative ambiguity. A clearer operational definition, perhaps supported by a table, would improve the clarity.

We thank the Referee for this raising this issue and apologize for any confusion caused by our previous brief operationalization. To address this point, we now include a new Figure in the manuscript to clarify the operationalizations (lines 170-176). While the specific input–output modalities vary across tasks, our primary aim was to compare inter-hemispheric (crossed) versus intra-hemispheric (uncrossed) trials, irrespective of the modality involved. We therefore believe that this terminology is valid and consistent with conventions in previous split-brain research. We hope that the addition of this Figure will make our definitions more transparent and help avoid any interpretative ambiguity.

Figure 3 Classic split-brain framework and task descriptions. **A.** Hemispheric organization of inputs and outputs. Stimuli in the left visual field (LVF) projects to the right hemisphere (RH), while stimuli in the right visual field (RVF) projects to the left hemisphere (LH). Motor outputs are also lateralized: the LH controls speech and the right hand, while the RH controls the left hand. **B.** Task summary table showing, for each task, the input and the response modality, and examples of uncrossed and crossed trial types. **C.** Examples illustrating uncrossed and crossed trials for each task.

10. Some tasks, such as the topognosis task, are insufficiently explained in the main text.

The manuscript has been reorganized to follow the journal's structure. In this format, the topognosis task is introduced earlier, and together with the new figure (see our response to Comment #9), we hope that the task and condition descriptions are now presented more clearly.

11. I recommend reporting time from surgery in months rather than “age at testing”.

We added a new column to Table 1, please see below.

Table 1 Clinical Profiles of Callosotomy Patients

Patient	Age at surgery	Age at testing	Time from surgery in months	Sex	Handedness	Intelligence	Extent of Callosotomy	Education (years)
BT*	22	28	70 ⁺	M	Right	Low average	Partial	10 years
TJ	49	52	34	F	Right	Average	Complete	13 years
LJ	57	60	26	M	Right	Low average	Complete	10 years
NR	18	18	6	M	Right	Moderately impaired	Complete	12 years

Table 1. Patients Overview. *Patient BT was first assessed 70 months after surgery, showing no disconnections; a full battery was conducted later (~87 months post-surgery), yielding the same results. Patient Intelligence classifications are based on the Stanford–Binet Fifth Edition (SB5) classification.^{21,22}

Reviewer 3 comments

1. Authors reported in methods that the anterior, posterior and hippocampal commissures remained intact in all patients. In the Introduction section, I suggest mentioning the role of the commissural auxiliary pathways when the corpus callosum is absent or altered in its anatomy or function. Commissural auxiliary pathway could support residual interhemispheric integration of visuospatial and sensorimotor functions (i.e. spatial orientation, visual processing, and motor coordination). Still, they cannot fully compensate for the absence of the corpus callosum. This provides relevant insight, considering that your results highlight that the preservation of the splenium appears to be the only difference among patients that may allow for inferences about its role in supporting reorganisation processes critical for interhemispheric functioning.

We agree that auxiliary commissural pathways (e.g., anterior, posterior, and hippocampal commissures) can indeed support limited residual inter-hemispheric integration. However, as noted by the referee, they are typically insufficient to fully compensate for the absence of the corpus callosum, which is why the classical disconnection syndrome still emerges for the patients with intact anterior commissure but severed corpus callosum (e.g., Gazzaniga et al., 1985, <https://pubmed.ncbi.nlm.nih.gov/4069368/>). To acknowledge this point, we have added a sentence in the Introduction (lines 59-62). We kept this addition concise, as these commissures were intact in all patients in our cohort and did not represent a source of variability between them:

“The corpus callosum (CC) is the largest white matter tract connecting the left and right cerebral hemispheres.¹ Evidence from callosotomy patients—in whom the CC is surgically severed—shows that the CC plays a crucial role in integrating sensorimotor events across the hemispheres. In lateralized tasks that isolate sensory input and motor output to a single hemisphere, these patients show a profound disruption in inter-hemispheric information flow, known as the *disconnection syndrome*. Although other commissural pathways—such as the anterior, posterior, and hippocampal commissures—typically remain intact and may provide limited residual inter-hemispheric integration², they cannot fully compensate for the loss of the corpus callosum, as patients still typically exhibit disconnection syndromes.³ For example, consider a callosotomy patient whose speech areas are located in the left hemisphere. When the word “hot” is presented in their left visual field (processed by the right hemisphere) and the word “dog” in their right visual field (processed by the left hemisphere), they verbally report seeing only the word “dog”. However, when asked to draw what they saw using their left hand (controlled by the right hemisphere), they draw a fire. Crucially, the emergent concept of a *hotdog* is never reported, even though each hemisphere accurately processes its respective input.”

2. The Methods and discussion sections are well-crafted, clearly and concisely written, with great detail and care. The entire manuscript is generally easy to follow for researchers and readers from a clinical background. A strong understanding of clinical assessment and evidence is undoubtedly necessary, yet the whole topic of disconnection syndrome remains complicated, and I appreciate how the authors have presented the entire topic. However, I suggest aligning the writing style of the Introduction and Results sections with that of the Method and Discussion, as this would elevate the overall quality of the manuscript. For example, lines 85-89 could be presented in a more effective way that would better highlight the originality of the study.

We thank the Referee for their positive assessment of the manuscript. We have revised the ending of our introduction, specifically we revised the introduction section on splenium-only cases to more directly emphasize the originality of our study (lines 93-96):

“...Only a few splenium-only cases exist in the literature, and their findings are mixed: some suggest the splenium supports broad inter-hemispheric integration,¹⁶ while others indicate it is insufficient beyond visual information.¹⁷ However, these studies have limitations—some lack MRI confirmation of exact preserved structures,¹⁶ while others conducted testing very soon after surgery (within 6 months), making it unclear whether any reorganization had occurred over time.¹⁷ As a result, the specific contribution of the splenium to inter-hemispheric integration remains unclear.

The current study addresses this gap by reporting on a rare, splenium-only callosotomy case, compared against three complete callosotomy patients.”

3. Table 1: Information regarding the participants’ education level may be relevant. Furthermore, I suggest including details on the participants’ sensory and motor integrity. While these aspects are likely reflected in the performance of the intrahemispheric tasks, explicitly reporting them would provide a more complete and informative clinical profile.

We appreciate the Referee’s helpful suggestion.

We have added a new “Education” column to Table 1:

Table 1 Clinical Profiles of Callosotomy Patients

Patient	Age at surgery	Age at testing	Time from surgery in months	Sex	Handedness	Intelligence	Extent of Callosotomy	Education (years)
BT*	22	28	70 ⁺	M	Right	Low average	Partial	10 years
TJ	49	52	34	F	Right	Average	Complete	13 years
LJ	57	60	26	M	Right	Low average	Complete	10 years
NR	18	18	6	M	Right	Moderately impaired	Complete	12 years

Table 1. Patients Overview. †Patient BT was first assessed 70 months after surgery, showing no disconnections; a full battery was conducted later (~87 months post-surgery), yielding the same results. Patient Intelligence classifications are based on the Stanford–Binet Fifth Edition (SB5) classification.^{21,22}

We have also added the following sentence to the Participants section under Methods (lines 131-132):

“All patients completed secondary school degrees (10-13 years of education)”

Next, we included the following section under Results (lines 338-342) to more explicitly report the sensory and motor integrity of the patients (as evidenced by the intra-hemispheric, uncrossed, trials):

“Across all intra-hemispheric (uncrossed) conditions, patients performed significantly above chance (overall $p < 0.05$; individual p -values per task are reported below). These results indicate intact sensory and motor integrity when tasks did not require inter-hemispheric information integration, whereas performance typically broke down once such integration was needed. Detailed results for each task are reported below.”

4. In the Results section and Table 2, the number of trials is reported with differences across patients and between intra- and interhemispheric tasks. In the Table 2 caption, the authors mention that this limitation regarding unmatched trial numbers would be discussed in the Results and Discussion sections. While it is possible to infer what they are referring to, and I understand the challenges involved in clinical assessments with patients, it is necessary to explain clearly why the number of trials differs among patients and between intra- and interhemispheric tasks.

We thank the Referee for highlighting this important aspect. We previously provided the reasons for trial number differences in each patient’s case description within the Results section, resulting in somewhat dispersed clarifications that likely confuse the reader. To rectify this issue, we have now added a short summary statement in the Results section (lines 343-349) that consolidates the main sources of variation:

“The number of trials varied across patients and tasks because of practical bedside constraints, including fatigue, limited testing time, and stopping some inter-hemispheric trials early to avoid frustration when patients struggled with task performance. For some tasks, when testing time was limited, we prioritized inter-hemispheric trials, once intra-hemispheric ones were performed easily, which led to uneven trial numbers. These case-specific reasons are detailed below;

nevertheless, the overall pattern of intact intra-hemispheric and impaired inter-hemispheric performance was generally consistent across patients.”

*In brief, trial numbers differed **between patients** due to individual factors. BT and NR (our younger patients) were able to complete testing with relatively balanced trial numbers, whereas TJ and LJ showed greater variability. LJ, as an older patient, had shorter testing sessions to minimize fatigue. TJ, who was highly verbal, presented more challenges. As also described in the Results section, during stereognosis she reported not feeling anything in her left hand; to avoid frustration we terminated this condition early (0/1) and instead administered additional topognosis trials (please also see our response to Comment #5 below) to ensure that she could actually “feel” her left hand. In the WISCR task, she struggled for an extended period with her right hand, offering lengthy explanations for why she could not perform the task; after two right-hand trials, which lasted quite long, we aborted the condition and switched to her left hand.*

*As for differences **between intra- and inter-hemispheric** trial numbers: as mentioned above, TJ was particularly challenging. In stereognosis and WISCR block design, her speaking left hemisphere often produced lengthy explanations for incorrect attempts, and these trials were therefore aborted early. In the hand gesture task (3/3 right hand vs. 4/12 left hand), because she completed the right-hand trials quickly and easily, we chose to allocate more of the limited testing time to the inter-hemispheric condition.*

5. Line 173-183: The authors’ interpretation of TJ’s perceptions appears reasonable and is likely to be broadly agreed upon. However, they might consider that her behaviour could appear unusual to readers who can only access limited information about her clinical history. Furthermore, Table 2, which reports her performance, raises several important considerations. In the task, the patient was asked to name everyday objects placed in her left hand, and she reportedly claimed to feel nothing in that hand. Is this accurate? Could this explain why TJ completed more trials than the other participants in the Topognosis task? I suggest including a few lines in the Discussion section addressing TJ’s behaviour. Additionally, it would be interesting to know how TJ performed in the control-mimicking task concerning her frustration and motivational issues.

When everyday objects were placed in TJ’s left hand, she reported feeling nothing. This is precisely why we conducted additional intra-hemispheric topognosis trials with her—to confirm that the issue was not due to sensory neglect or sensory loss in the left hand. We greatly appreciate the Referee’s careful observation and agree that our manuscript will benefit from clarifying this point more explicitly in the Results section (lines 379-381)

“When an object was placed in her left hand—processed by the mute right hemisphere—her disconnected speaking left hemisphere repeatedly insisted she felt

nothing. However, note that, during the tactile localization task reported above, she performed perfectly fine with her left hand—accurately locating touch stimuli within her left hand in the intra-manual trials. We administered a higher number of trials in the tactile localization (Topognosis) task for TJ compared to the other patients, specifically to confirm that she was able to perceive and localize the tactile input on the left hand—which she did. “

Regarding the control-mimicking task: TJ actually successfully completed 9 out of 9 trials with her left hand indicating no motor or motivational issues (Results section lines 415-420):

“TJ showed difficulty performing hand gestures with her left hand (4/12) when instructed verbally. Remarkably, she easily and accurately performed all gestures when mimicking the experimenter’s hand movements with her left hand (9/9), showing no signs of motor or motivational deficits. The difference between her left-hand performance following verbal commands (4/12) and when simply mimicking the experimenter’s hand gestures (9/9) was significant (observed difference in proportion correct = 0.66, $p = 0.004$)”

6. Lines 208-213. The interpretation provided by the authors regarding NR’s performance is compelling and could be widely accepted. However, the authors should consider whether any clinical information is available regarding the patient's performance on that specific task within the six months following surgery. For instance, if the patient performed the task shortly after surgery with a low success rate but achieved 100% success at your six-month evaluation, it might be appropriate to discuss the possibility of a learning effect. Conversely, if the patient already performed well from the outset, this would support the bilateral representation for the visuospatial processing hypothesis. The successful performance guided by “Regular rehabilitation exercises involving similar tasks” reflects a generalisation of the task, an excellent outcome 6 months after surgery. The patient is young and his brain plasticity certainly played a role but – to entertain a conversation with the authors - Might this explanation be considered somewhat weak, given the specificity of the WAIS Block Design Test according to your experience? If previous data on the Block Design Test or other neuropsychological assessments—such as memory tasks—were available, they would provide substantial added value to the interpretation of the findings.

We thank the Referee for raising this important point and for the thoughtful question. Unfortunately, we do not have any immediate post-operative assessments for any of the patients in this cohort outside of an ultra-acute, post-surgical period. We acknowledge this limitation for BT (see our response to Reviewer 1, comment 2). The present manuscript is therefore focused on stable outcomes obtained >6 months after surgery,

contrasting the partial and complete callosotomy patients. A more detailed characterization of immediate versus longer-term changes, including potential recovery, plasticity, or rehabilitation effects, will require a dedicated longitudinal study, which we aim to pursue in a future work.

Regarding the possibility of generalization of the learning effect from rehabilitation, we unfortunately do not have details about the specific exercises NR completed. We agree with the Referee that the WISC-R Block Design task is relatively specific.

For example, classic split-brain work has shown that patients can recognize and discriminate the very same visuospatial patterns when tested in a non-construction format (e.g., choosing the correct pattern from among alternatives after tachistoscopic presentation). Both hemispheres typically succeed in such perceptual pattern-matching tasks. Thus, the deficit appears not to involve perceptual pattern recognition per-se, but rather the manipulation and reconstruction of patterns in 3D space using cubes—a process Gazzaniga and colleagues referred to as “manipulo-spatial” (LeDoux et al., 1977, <https://pubmed.ncbi.nlm.nih.gov/600369/>).

7. Bayne 2008 doesn't seem to be mentioned in the text but only reported in the reference list. I suggest checking and revising citations' sequential numbering to meet the journal's standards before reaching the final stages.

We apologize for this error and have fixed this oversight.

8. Figure 4 and the discussion section deserve a dedicated appreciation: In these branches of clinical research, current and future studies can only benefit from past single-case reports, as they provide valuable points of comparison and help strengthen the evidence base on the topic.

We thank the Referee for this positive and encouraging comment! We fully agree that single-case reports are invaluable, as they provide insights that can reshape perspectives in the field and serve as essential reference points both historically and in contemporary clinical and cognitive neuroscience research. We hope our work will contribute to the field by introducing this rare splenium-only case along with this new cohort of complete callosotomy cases, thereby stimulating future research and discussion on how the brain may adapt to perturbation through residual pathways.

Replies to all Referees

Overview

We thank all three Referees for re-evaluating our manuscript and for their positive and encouraging feedback. We are pleased that the revisions have addressed the main concerns raised in the first round. Below, we respond point-by-point to the remaining comments.

Reviewer 1 comments

Dear authors, I have carefully read the revised manuscript. Thank you very much for your efforts in addressing my concerns. In my opinion, the manuscript is now more nuanced, and the limitations clearly acknowledged. I have no further comments. Congratulations on the excellent work.

We thank the Referee for the positive evaluation and encouraging feedback

Reviewer 2 comments

I would like to thank the authors for the clarifications provided. The manuscript has substantially improved. The addition of the neuropsychological summary table (which could be included as supplementary material) and the new figure are both welcome and help clarify the results. I only have one remaining minor comment on the methods of the DWI acquisition. Although the tract image is presented primarily for visualisation purposes, the manuscript still needs to include the essential diffusion acquisition details and a brief description of how the tracts were obtained. If the authors prefer not to expand the main text, this could be briefly stated in the Methods section with reference to supplementary material for full details. Showing the tractography without reporting how these tracts were derived leaves a methodological gap that should be clarified. Once this point is addressed, I believe the manuscript will be suitable for publication.

We thank the Referee for the positive and constructive feedback. We did not include the neuropsychological summary table as a supplementary material since the relevant information is already provided in the main text, and there are no other supplementary materials accompanying the manuscript. However, as suggested, we have now added a detailed description of the DWI acquisition and tractography details to the Methods section to clarify how the tract visualizations in Figure 2 were obtained. The new text is provided below (lines 294-311 in the main manuscript file).

Diffusion MRI Acquisition and Fiber Tractography

To generate tractography images for illustration purposes (**Fig 2**), diffusion-weighted images were acquired on a 3 T Siemens Vida at the Bethel Epilepsy Center using a 32-channel head coil. A single-shot echo-planar imaging sequence was used with one $b = 0$ volume, and diffusion-encoding gradients applied along 30 directions at $b = 1000$ s/mm² (TR = 13.2 s, TE = 83 ms, flip angle = 90°, slice thickness = 2 mm, voxel

size = $2 \times 2 \times 2$ mm³, FOV = 256 mm). Parallel imaging was applied using GRAPPA (acceleration factor = 2).

Tractography was performed in DSI Studio.³³ Diffusion data were reconstructed using generalized q-sampling imaging (GQI),³⁴ and deterministic streamline tracking was applied.³⁵ The anisotropy threshold was randomly selected. The step size was set to voxel spacing. Parameters such as the angular threshold (35-90° range), and the total number of seeds were placed (around 50000 seeds) were individually adjusted across patients to achieve optimal anatomical visualization. Tracks shorter than 40 mm or longer than 200 mm were excluded. Tractography was used solely for illustrative purposes in Figure 2 and was not analyzed quantitatively. Only select commissural, cerebellar, and brainstem fibers were retained for display after manual pruning to enhance clarity. Each tractography rendering was overlaid on the patient's own T1-weighted structural image to highlight the anatomical landmarks relevant to surgical outcomes.

Reviewer 3 comments

I approve the authors' responses and the competence with which they have provided them. I appreciated the authors' clinical experience in addressing the issues raised by the reviewers, as well as in providing specific answers in addition to the direct changes in the manuscript. This work meets the journal's standards and contributes to promoting future research and discussion on how the brain may adapt to perturbation through residual pathways.

We thank the Referee for the thoughtful and encouraging comments.